# CORE: Reducing UI Exposure in Mobile Agents via Collaboration Between Cloud and Local LLMs

Gucongcong Fan[1]    Chaoyue Niu[1][*]    Chengfei Lyu[2]    Fan Wu[1]    Guihai Chen[1]

[1]Shanghai Jiao Tong University    [2]Alibaba Group

## Abstract

Mobile agents rely on Large Language Models (LLMs) to plan and execute tasks on smartphone user interfaces (UIs). While cloud-based LLMs achieve high task accuracy, they require uploading the full UI state at every step, exposing unnecessary and often irrelevant information. In contrast, local LLMs avoid UI uploads but suffer from limited capacity, resulting in lower task success rates. We propose **CORE**, a **CO**llaborative framework that combines the strengths of cloud and local LLMs to **R**educe UI **E**xposure, while maintaining task accuracy for mobile agents. CORE comprises three key components: (1) **Layout-aware block partitioning**, which groups semantically related UI elements based on the XML screen hierarchy; (2) **Co-planning**, where local and cloud LLMs collaboratively identify the current sub-task; and (3) **Co-decision-making**, where the local LLM ranks relevant UI blocks, and the cloud LLM selects specific UI elements within the top-ranked block. CORE further introduces a multi-round accumulation mechanism to mitigate local misjudgment or limited context. Experiments across diverse mobile apps and tasks show that CORE reduces UI exposure by up to 55.6% while maintaining task success rates slightly below cloud-only agents, effectively mitigating unnecessary privacy exposure to the cloud.[2]

## 1 Introduction

Task automation on smartphones aims to enable a mobile agent to autonomously execute tasks based on user-provided descriptions, thereby freeing the user's hands and enhancing the user experience. Recent studies have introduced various mobile agents driven by large language models (LLMs) [31, 36, 33, 32, 46, 14]. These agents operate in a human-like manner. Typically, they receive user commands, sequentially assess the page state, generate the corresponding sub-task, and determine how to interact with the graphical user interface (GUI) until the task is completed.

In practice, most agents rely on cloud-based proprietary LLMs [31, 36, 14], such as GPT-4o accessed via OpenAI's paid API. These models contain at least hundreds of billions of parameters and demonstrate powerful reasoning capabilities, enabling mobile agents to achieve high success rates on a variety of simple tasks. Despite the promising results, several limitations remain. In mobile automation tasks, the agent must capture the page information at every step and upload it to the cloud-based LLM. We find that the information uploaded for decision-making is excessive, with only a small portion being relevant and the rest redundant. For many sub-tasks executed on smartphone interfaces, the only required action is to click a functional button, which relies solely on the local information of specific UI elements rather than on the full page context. However, current mobile agents do not take advantage of this property, they routinely upload the complete page to the cloud-based LLM, thereby transmitting a large amount of redundant information that may even contain sensitive user data. As shown in Figure 1, if the user query is to search something, only the search

---

[*]Chaoyue Niu is the corresponding author (rvince@sjtu.edu.cn).

[2]The code is available at `https://github.com/Entropy-Fighter/CORE`.

39th Conference on Neural Information Processing Systems (NeurIPS 2025).

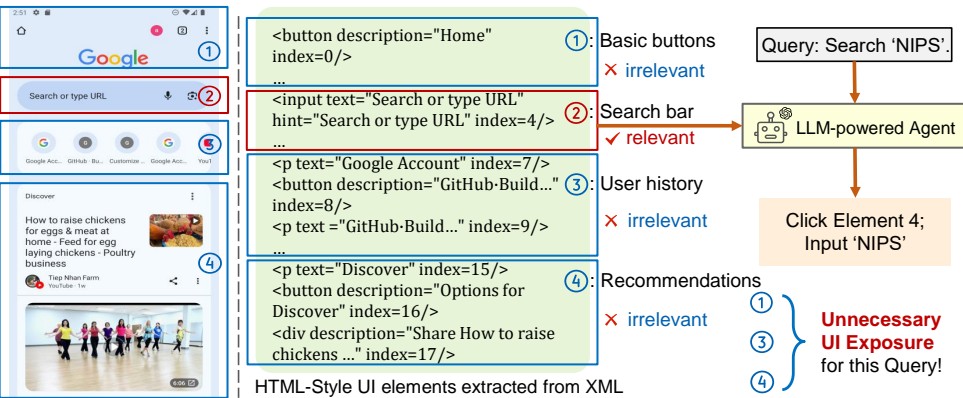

Figure 1: Given the user query "Search NIPS", only the search bar on the Chrome start page is task-relevant. Uploading just this block to the agent is sufficient for correct decision-making, indicating that other UI blocks (basic buttons, browsing history, recommendations) are unnecessary exposure.

bar needs to be activated and other information on the page is superfluous. Smartphones have a privacy-sensitive nature and users are reluctant to expose their private information too much.

One seemingly straightforward solution is to deploy open-source LLMs locally, thereby avoiding any data exposure to the cloud. However, these models typically contain only billion-scale parameters and thus exhibit only elementary reasoning capabilities. Their performance on mobile automation tasks is poor, yielding low success rates and a substantial performance gap compared to cloud LLMs.

To solve these problems, we propose **CORE**, a collaborative architecture that leverages both cloud-based strong LLMs and locally deployable light-weight LLMs. First, we employ a hierarchy-guided strategy based on the XML tree structure to partition the page into structured blocks. Instead of uniform segmentation that disrupts inherent UI structures, the partitioning follows the original UI design intent and preserves logical element clusters. The next step is to locally rank the blocks by their importance to the current sub-task. The local LLM's limited planning capability leads to inaccurate sub-task generation, which compromises the validity of the ranking. To remedy this, we design a collaborative planning module that combines the cloud LLM's powerful planning ability with the local model's basic reasoning capability and its advantage of full-page access. Specifically, the local LLM generates multiple candidate sub-tasks from different blocks. The cloud model, by comprehensively analyzing these candidates, indirectly infers the page context and selects the most plausible sub-task or generates a more accurate one. We further design a collaborative decision module that integrates the coarse-grained filtering of the local LLM with the fine-grained decision-making of the cloud LLM. The confirmed sub-task is fed back to the local LLM to guide block ranking. We pass the top-ranked block to the cloud LLM for element-wise decision-making. We also introduce a multi-round accumulation mechanism. If the cloud LLM considers the selected block insufficient for a proper decision, it will incrementally ask the local LLM for additional blocks until the provided information is judged to be adequate for a confident decision. This collaborative process achieves a dynamic balance between information sufficiency and reduction. These designs form a novel asymmetric collaborative framework, different from prior multi-agent approaches that typically employed cloud LLMs with distinct roles under full UI access. CORE establishes collaboration between a strong cloud LLM without direct UI access and a weak local LLM with full UI visibility but limited reasoning capability, coordinating by leveraging their complementary strengths.

Our main contributions are as follows: (1) To the best of our knowledge, we are the first to study the problem of reducing unnecessary information exposure and uploading only minimal sufficient UI contents to cloud-based LLMs during mobile agent's decision steps; (2) we propose an XML-based layout-aware block partitioning method to preprocess UI content, and a collaborative framework combining local and cloud LLMs for joint planning and decision-making, reducing UI exposure to the cloud while preserving decision accuracy; and (3) on the public DroidTask dataset (143 tasks, 12 apps) [36], our GPT-4o + Gemma2-9B setup reduces UI exposure by 55.6% with task success rate just 4.9% lower than GPT-4o. On the public AndroidLab dataset (98 tasks, 9 apps) [40], GPT-4o + Qwen2.5-7B achieves a 29.97% reduction with only a 3.06% drop, showing practical effectiveness. Moreover, CORE significantly reduces sensitive information exposure, lowering the number of sensitive UI elements uploaded to the cloud by 70.49% on DroidTask and 38.84% on AndroidLab.

## 2 Problem Formulation

We first introduce the workflow of an XML-based mobile agent, which is a sequential process to complete a user's given task. At each time step $t$, the agent receives the following inputs:

**Task description** $T \in \mathcal{T}$: a natural language instruction given by the user at the beginning $t = 0$, e.g., "Set an alarm for 8 AM".
**Decision history** $H_t \in \mathcal{H}$: a sequence of past decisions executed on-device up to step $t$, e.g., [LaunchApp Clock, Click `<button text="Add" description="Add alarm" index=2/>`].
**Current UI state** $X_t \in \mathcal{X}$: a set of HTML-style UI elements, parsed from the current XML file, including attributes like `text`, `content-description`, `resource-id`, etc. e.g., [`<p text="Clock" index=0/>`, `<button id="settings" description="Settings" index=1/>`, ...]

The LLM processes these, internally determines an intermediate sub-task $Z_t$ to be executed on the current page, and then makes the decision based on it:

**Decision** $D_t \in \mathcal{D}$: a tuple of the target UI element $E_t$ and the interaction type $A_t$ (with the action space typically including click, long press, input text and scroll).

We use $\mathcal{T}$, $\mathcal{H}$, $\mathcal{X}$ and $\mathcal{D}$ to denote the spaces of task descriptions, decision histories, UI states and decisions respectively. The LLM's decision function is defined as

$$f : \mathcal{T} \times \mathcal{H} \times \mathcal{X} \to \mathcal{D}. \tag{1}$$

The agent executes the decision $D_t$ on mobile device. Then, the UI state is updated to $X_{t+1}$ and $D_t$ is appended to $H_t$ to form the updated decision history $H_{t+1}$. This iterative process continues until the LLM determines that the task is complete, at which point the agent terminates further execution.

Building upon the workflow described above, we aim to develop a strategy that selects a subset $X_t'$ from the current page input $X_t$, thereby reducing the volume of uploaded data while preserving decision quality. In other words, the objective is to discard redundant information that is unnecessary for the task at hand, ensuring that the decision outcome remains largely consistent with that obtained using the full page data. Formally, we seek a selection strategy $\mathcal{S}$ such that

$$X_t' = \mathcal{S}(T, H_t, X_t) \subseteq X_t. \tag{2}$$

The proposed strategy must satisfy two key objectives. First, it should minimize the superfluous information on the current page, i.e., make $|X_t'|$ (the cardinality of $X_t'$) as small as possible. Second, it must ensure consistency in the decision-making process before and after data reduction, that is,

$$D_t = f(T, H_t, X_t) = D_t' = f(T, H_t, X_t'), \tag{3}$$

which implies that, the reduction of information should not affect the decision outcome. In practice, a small proportion of deviation is acceptable, but the decisions should remain nearly identical.

Thus, the overall optimization problem is formulated as:

$$\min_{\mathcal{S}} \; \mathbb{E}_{(T,H_t,X_t)} \left[|X_t'|\right] \quad \text{s.t.} \quad \mathbb{E}_{(T,H_t,X_t)} \left[\mathbb{I}\Big(f(T, H_t, X_t) \neq f(T, H_t, X_t')\Big)\right] \leq \varepsilon. \tag{4}$$

Here, $\mathbb{I}(\cdot)$ denotes the indicator function, and the expectation is taken over the distributions of task descriptions, decision histories and page inputs, thereby allowing for a controlled degree of deviation in the decision outcomes, with $\varepsilon$ representing a small threshold that bounds such deviation. This formulation encapsulates the dual challenges of minimizing unnecessary data while ensuring minimal deviation in decision outcomes.

## 3 CORE Design

The design goal is to upload only sufficient UI content to the cloud, minimizing unnecessary information exposure while maintaining task performance. To this end, we propose a collaborative framework CORE that leverages the cloud LLM's strong reasoning capabilities and the local LLM's full access to the UI context. CORE consists of three stages: the *on-device preprocessing stage* takes the XML screen as input and divides UI elements into semantically coherent blocks $B_t$; the *collaborative planning stage* takes the task $T$, history $H_t$, and blocks $B_t$ (accessible only to the local LLM) to determine the current sub-task $s^*$; and the *collaborative decision-making stage* takes $T$, $H_t$, $s^*$, and subset of $B_t$ to output a device-executable decision $D_t$, specifying the target UI element and action. Figure 2 illustrates the pipeline; further details are provided below.

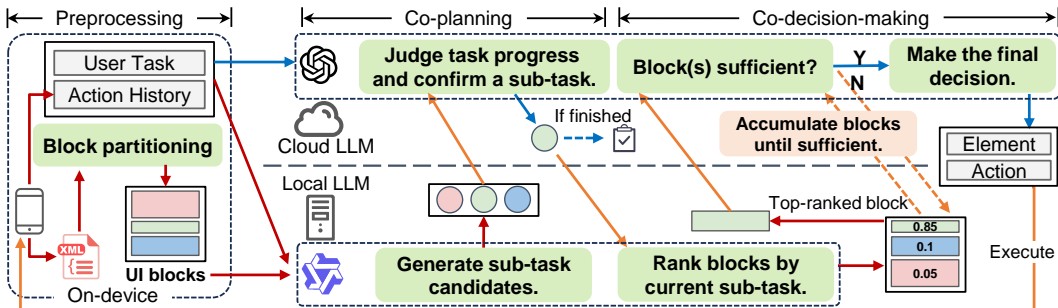

Figure 2: Overview of our **CORE** pipeline. It has three stages: on-device pre-processing, co-planning, and co-decision-making. In the latter two stages, it leverages the cloud LLM's strength of strong reasoning abilities and the local LLM's strength of full access to the UI blocks to collaborate.

## 3.1 Layout-aware Block Partitioning

Current XML-based mobile agents convert the screen represented by XML file into a simplified HTML-style UI elements list $X_t$ so that LLMs can better understand, with each element like <button text="Add" description="Add alarm" index=2>. The drawbacks are that the list of elements becomes lengthy on complex screens, and the structural relationships implied by the original XML layout are ignored. Actually, these UI elements can be naturally grouped into several coarse-grained blocks based on the XML tree structure. Therefore, we propose a layout-aware block partitioning strategy.

A depth-first search is applied to extract important UI elements (e.g. clickable and containing some semantics, editable) from the XML tree, which follows the traditional processing. However, unlike prior works, during traversal, we assign each node a unique index and record the ancestor path $A(x) = [a_0, a_1, \ldots, a_{k-1}]$ of every important node $x \in X_t$, where $a_0$ is the root's index and $a_{k-1}$ is the index of $x$'s parent. Many unimportant nodes skipped during traditional processing (e.g., layout containers like FrameLayout) act as shared ancestors of multiple important elements. The elements under the same ancestor container node are more likely to belong to the same logical or visual region due to the UI layout design. Given all ancestor paths $\{A(x)\}$, we traverse the ancestor levels from depth $i = 0$ up to the maximum path length $k_{\max} - 1$. At each level $i$, we partition important elements into different blocks by their different $i$-th ancestor $a_i$, where elements under the same $a_i$ are grouped in the same block. This produces block partitions of varying granularity, coarser at shallower levels (e.g., $i = 0$ yields a single block), and finer at deeper levels (e.g., $i = k_{\max} - 1$ yields the most blocks). We select the first level $i$ where the partition results in at least 3 distinct blocks, and return the blocks as the final layout-aware grouping. A toy example is shown in Figure 3.[3]

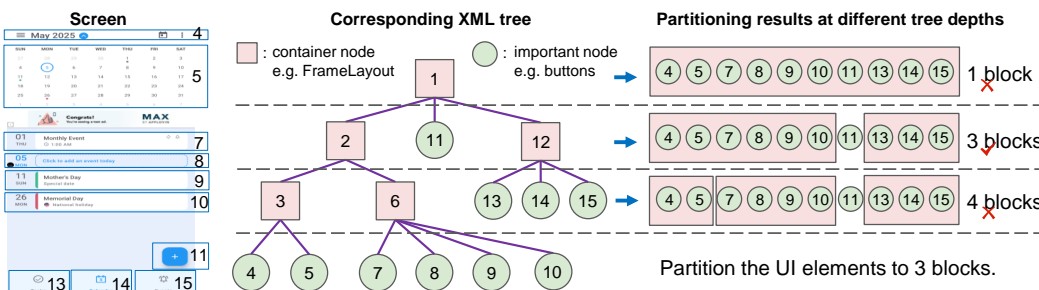

Figure 3: Our layout-ware block partitioning on a Calendar page. By comparing the ancestors of all important nodes at the same tree depth, we group them into blocks. Different depths yield different partition granularities; we select a suitable one to obtain 3 coherent blocks.

## 3.2 Collaborative Planning

The local LLM uses its access to full UI blocks to convey page understanding to the cloud LLM, which then uses superior planning skills to generate more accurate sub-tasks. We adopt a divide-and-

---

[3]We simplify the XML tree in Figure 3 for illustration by removing or merging some nodes.

conquer strategy to make local LLM predict the possible sub-task candidate for each UI block, as it is limited in long-context processing. For each block, we input the user-provided task description, the historical operations, and itself into the local LLM. It is required to infer and output the most reasonable sub-task that could be executed within the block without any leakage of direct element information. After processing all blocks, we aggregate the sub-tasks generated by the local LLM into a candidate set. These candidates, together with the global task description and historical operations, are then sent to the cloud LLM. The cloud LLM selects the most suitable candidate as the current sub-task, or revises/generates one if none of them is appropriate.

The sub-task candidate derived from each block represents high-level abstractions over the block's UI elements, capturing what the local LLM considers the most important for that block. As the blocks are partitioned according to the UI's internal logical structure, the cloud LLM can infer the rough functionalities of different blocks based on the candidate sub-tasks without access to detailed UI elements. Through comprehensive analysis of the candidate sub-tasks, the cloud LLM can piece together an approximate understanding of the overall page layout and content. This indirect perception enables the cloud LLM to make more accurate judgments about the task progress and ensures its planning remains grounded in the actual UI context.

### 3.3 Collaborative Decision-making

We provide the local LLM with the full set of page blocks along with the confirmed sub-task given by co-planning, and ask it to assign a score to each block based on its estimated importance for executing the sub-task. We adopt a probabilistic scoring scheme, where the local model predicts the probability that each block is useful for the current sub-task, and the scores are normalized to sum to one. This scoring produces a ranking over all blocks, reflecting the local LLM's belief about their relevance to the sub-task at the block level. The block with the highest score which is deemed most relevant and useful by the local model is then transmitted to the cloud LLM. The cloud LLM, with its stronger fine-grained reasoning capabilities, performs element-level decision-making: selecting the specific UI element to interact with and determining the action type.

We also introduce a multi-round accumulation mechanism to handle challenging tasks requiring long context and offers fault tolerance against local filtering errors. Upon inspecting the initially provided block, the cloud LLM evaluates whether the available information is sufficient for a confident decision. If not, it requests additional blocks from the local model. The local model then selects the next most probable block among those not yet uploaded and sends it to the cloud side. The cloud model incrementally integrates the newly received block with previously accumulated information and re-evaluates the decision. This iterative process continues until the cloud model judges that it has gathered enough context to make an informed and reliable decision. By dynamically adjusting the amount of information retrieved, this mechanism strikes a balance between minimizing information upload and ensuring decision accuracy. Even if the local model initially mis-ranks blocks or provides insufficient context, the cloud model can progressively recover the necessary information through incremental retrieval, maintaining robustness against local errors and variations in task difficulty.

Algorithm 1 summarizes the collaboration mechanism between the cloud and local models, as described in Sections 3.2 and 3.3.

## 4 Evaluation

### 4.1 Evaluation Setup

**Datasets.** We use two benchmark datasets for mobile agents: **DroidTask** [36] and **AndroidLab** [40]. The DroidTask dataset consists of 158 tasks across 13 basic apps such as Calendar, Clock, Dialer, and others. After filtering out tasks with compatibility issues, we obtain 143 tasks from 12 compatible apps. Some task instructions are revised for clarity while preserving their original intent. The

---

**Algorithm 1:** Cloud-local collaboration at Step $t$

**Input:** Task $T$, History $H_t = [D_1, ..., D_{t-1}]$, UI blocks $B_t = [b_1, ..., b_k]$

**Output:** Decision $D_t$ specifying action $a$ and target element $e$

$S \leftarrow [\,], C \leftarrow [\,]$;
**foreach** $b_i$ *in* $B_t$ **do**
$\quad s_i \leftarrow \text{LLM}_{\text{local}}.\text{generateSubtask}(T, H_t, b_i)$;
$\quad$ Append $s_i$ to $S$;
$s^* \leftarrow \text{LLM}_{\text{cloud}}.\text{confirmBestSubtask}(T, H_t, S)$;
$B_{\text{ranked}} \leftarrow \text{LLM}_{\text{local}}.\text{rank}(B_t, s^*)$;
**foreach** $b$ *in* $B_{ranked}$ **do**
$\quad$ Append $b$ to $C$;
$\quad (a, e) \leftarrow \text{LLM}_{\text{cloud}}.\text{makeDecision}(T, H_t, C)$;
$\quad$ **if** $(a, e) \neq$ *None* **then**
$\quad\quad$ **return** $(a, e)$
**return** *None*

---

AndroidLab dataset contains more challenging tasks in apps with more complex UI layuouts, such as Maps, Music, and Zoom. We take all the 98 operation tasks across 9 apps.

**Metrics.** Our evaluation focuses on three key aspects: task accuracy, UI exposure reduction, and the costs. These are measured by the following metrics.

*Task Success Rate.* We strictly follow the official evaluation criteria provided by each dataset to assess task completion. For Droidtask, a task is considered successful if the human-annotate action sequence is a subsequence of the actions executed by the agent. For AndroidLab, each task defines one or more key states that must be reached or triggered for successful completion. We extract key UI elements from these key states and check whether the agent's execution process covers these elements in the XML of any visited screen. A task is marked as successful if all required key elements are observed during execution. The success rate is the percentage of successfully completed tasks.

*Reduction Rate.* We measure the reduction in UI elements uploaded to the cloud compared to a GPT-4o baseline. To ensure fairness, we consider only the rounds where both methods make the same decision on the same UI screen. Let $E_{\text{GPT-4o}}$ denote the number of UI elements required by the GPT-4o baseline, and let $E_{\text{ours}}$ denote the number required by our design under the same condition. The reduction rate is calculated as:

$$\text{Reduction Rate} = \frac{E_{\text{GPT-4o}} - E_{\text{ours}}}{E_{\text{GPT-4o}}}. \tag{5}$$

In addition to evaluating all uploaded elements, the same reduction metric is also computed specifically for sensitive elements, whose sensitivity is determined by another LLM, Qwen2.5-Max. For completeness, we also provide other reduction metrics under alternative comparison settings in the Appendix E.

*Latency and Cloud LLM Token Usage.* Since the overall latency bottleneck lies in the reasoning phase rather than in action execution, we report the total inference time of LLM. For our design, which integrates both local and cloud-based LLMs, we provide separate inference latency for each. To reflect the associated financial cost, we also report the total token usage, including both input and output tokens, for GPT-4o API calls.

**Baselines.** To demonstrate our design's balance between task accuracy and UI exposure reduction, as well as the necessity of cloud–local collaboration, we compare against the following baselines.

*Cloud-Only Baseline.* This baseline follows the mainstream workflow of mobile agent, like AutoDroid [36], as shown in Section 2, and is powered solely by a cloud-based LLM (e.g., GPT-4o), benefiting from strong reasoning capabilities and typically achieving high task success rates. However, the full UI needs to be uploaded to the cloud at every step. Built on full UI context, this baseline serves as the upper bound for task accuracy, but comes with the highest UI exposure.

*Local-Only Baseline.* This baseline calls only a locally deployable LLM and does not transmit UI to the cloud, which serves as the upper bound for UI exposure reduction (i.e., 100%), but suffers from significantly lower task success rate.

*Basic-Order Ranking.* It is a variant of our design by replacing the local LLM's intelligent block ranking with ranking UI blocks in a fixed order (e.g., top to bottom or left to right), mimicking natural human reading behavior. The cloud LLM sequentially receives and accumulates blocks in this basic order until it can make a confident decision. This baseline reflects a trivial rule-based filtering strategy that does not involve local LLM.

*Random Ranking.* It is another variant of our design by ranking and sending UI blocks in a random order to the cloud.

**Implementation Details.** For the cloud LLM, we use GPT-4o-2024-11-20 [12] accessed via OpenAI's official API. For the local LLMs, we deploy quantized versions of Gemma2-9B-Instruct [30], Qwen2.5-7B-Instruct [41], and LLaMA3.1-8B-Instruct [9] using Ollama [23], running on a consumer-grade NVIDIA GeForce RTX 4090D GPU. For mobile agent execution environments, we use a real Honor Play3 smartphone with Android 9 for the Droidtask dataset, and the official Pixel 7 Pro emulator with Android 13 from AndroidLab for its dataset. UI states are extracted as XML files using uiautomator2 [5] for decision making, and actions are executed in the mobile environment via Android Debug Bridge (ADB) [7]. For both cloud-only and local-only baselines, we adopt the same prompt configurations as used in AutoDroid [36].

## 4.2 Main Results and Analysis

Table 1: Our design vs. Baselines from task success rate and UI exposure reduction rate.

| Methods | DroidTask Dataset | | AndroidLab Dataset | |
|---|---|---|---|---|
| | Success Rate | Reduction Rate | Success Rate | Reduction Rate |
| GPT-4o Baseline | **74.13%** | 0.00% | **44.90%** | 0.00% |
| Gemma 2-9B Baseline | 32.87% | **100.00%** | 17.35% | **100.00%** |
| Qwen 2.5-7B Baseline | 14.69% | **100.00%** | 5.10% | **100.00%** |
| LLaMA 3.1-8B Baseline | 9.79% | **100.00%** | 11.22% | **100.00%** |
| Ours (GPT-4o + Gemma 2-9B) | **69.23%** | 55.60% | 37.76% | **34.96%** |
| Ours (GPT-4o + Qwen 2.5-7B) | 61.54% | 41.89% | **41.84%** | 29.97% |
| Ours (GPT-4o + LLaMA 3.1-8B) | 49.65% | 40.56% | 34.69% | 31.65% |

**Comparisons with Baselines.** As shown by Table 1, the GPT-4o baseline achieves the highest success rate with 0% UI exposure reduction, while local-only baselines perform significantly worse but achieve 100% reduction by avoiding any cloud uploads. Our design strikes a balance between these baselines. For the Droidtask dataset, the best 4o & Gemma combination achieves a 55.6% reduction with only a 4.9% drop in success rate compared to the GPT-4o baseline, while outperforming the Gemma-only baseline by 36.36%. For the AndroidLab dataset, the best-performing 4o & Qwen combination achieves a small 3.06% drop in success rate compared to GPT-4o, and outperforms the Qwen-only baseline by 36.74%, with a 29.97% UI reduction. Our best reduction-focused setting (4o & Gemma) achieves a 34.96% exposure reduction, with a 7.14% drop in success rate, still outperforming the Gemma-only baseline by 20.41%. Our design combines the strengths of both cloud and local LLMs, selectively transmitting only essential UI elements to the cloud, thereby reducing data exposure while maintaining comparable performance to the cloud-only baseline.

**Quantitative Privacy Analysis of Sensitive UI Exposure** To quantitatively evaluate the privacy benefits of our framework, we conducted a detailed analysis of the sensitive information omitted by CORE, comparing the sensitive UI elements uploaded to the cloud by the GPT-4o baseline and by our method (under the configuration GPT-4o & Gemma 2-9B). We categorized sensitive data into eight major types: (1) Identity & Account (e.g., username, profile); (2) Location & Schedule (e.g., home address, calendar events); (3) Contacts & Communication (e.g., contact information, messages, call logs); (4) Media & Files (e.g., file name, file content); (5) Device & Usage Info (e.g., device ID, storage); (6) Behavior & Preferences (e.g., interests, browsing history, custom settings); (7) Finance & Security (e.g., payments, passwords, transactions); and (8) Other Sensitive Information. For each task step, all uploaded UI elements were analyzed using Qwen2.5-max to identify sensitive content and assign each element to the corresponding category.

Table 2: Comparison of uploaded sensitive UI elements between the GPT-4o baseline and our CORE under two datasets.

| Category | DroidTask Dataset | | | AndroidLab Dataset | | |
|---|---|---|---|---|---|---|
| | GPT-4o | CORE | Reduction | GPT-4o | CORE | Reduction |
| Identity & Account | 91 | 50 | 45.05% | 189 | 111 | 41.27% |
| Location & Schedule | 226 | 70 | 69.03% | 299 | 114 | 61.87% |
| Contacts & Communication | 458 | 122 | 73.36% | 273 | 203 | 25.64% |
| Media & Files | 147 | 32 | 78.23% | 12 | 12 | 0.00% |
| Device & Usage Info | 0 | 0 | / | 10 | 5 | 50.00% |
| Behavior & Preferences | 45 | 10 | 77.78% | 77 | 53 | 31.17% |
| Finance & Security | 2 | 2 | 0.00% | 102 | 90 | 11.76% |
| Other Sensitive Information | 0 | 0 | / | 1 | 1 | 0.00% |
| **Total** | **969** | **286** | **70.49%** | **963** | **589** | **38.84%** |

The results in Table 2 show that CORE significantly reduces the number of sensitive UI elements uploaded to the cloud by 70.49% on DroidTask and 38.84% on AndroidLab. These reductions are consistent with the overall element reduction rates (55.60% and 34.96%, respectively) reported in the main text, confirming that reducing overall UI exposure effectively lowers privacy risks.

Further manual inspection was conducted to understand the sources of reduction. The key findings indicate that most sensitive data fall into the Location & Schedule and Contacts & Communication categories, primarily due to apps such as Calendar, Contacts, and Messenger. Representative cases of the CORE framework are summarized below.

- In a Calendar task (creating a new event), only the UI context relevant to the new event is uploaded, while previously scheduled events on the screen which may reveal the user's calendar are omitted.
- In a Contacts task (updating a phone number), only the phone number field is transmitted, whereas unrelated fields such as email and birthday are excluded.
- In a Messenger task (sending a message), only the current message block is uploaded to the cloud, with prior chat history kept local.

These cases demonstrate that CORE effectively reduces unnecessary sensitive UI exposure, thereby enhancing user privacy, comfort, and trust in mobile agent interactions.

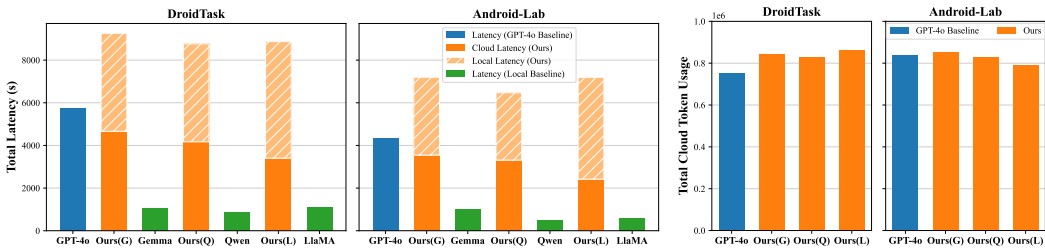

Figure 4: Our design vs. Baselines from latency and cloud LLM token usage.

**Cost Analysis.** Figure 4 shows a detailed comparison of latency and cloud LLM token usage between our design with different local LLMs and the baselines. The local baselines yield the lowest latency, but perform too poorly to be considered. Our design reduces cloud inference time by up to 44% compared to the GPT-4o baseline, because the cloud LLM only needs to deal with the filtered UI content. However, this cloud-local collaborative framework incurs additional overhead on the local side, since the local LLM handles sub-task generation and content filtering. Consequently, the overall latency of our method is $1.52\sim1.61\times$ and $1.49\sim1.66\times$ that of the GPT-4o baseline on DroidTask and AndroidLab, respectively. Regarding cloud LLM token usage, our method consumes $1.10\sim1.15\times$ and $0.94\sim1.01\times$ the tokens of the GPT-4o baseline on the two datasets respectively. Although we substantially reduces the elements uploaded to the cloud, the cloud LLM still incurs token overhead from co-planning and multi-round interactions with the local LLM to accumulate sufficient blocks for co-decision-making. Notably, on the more complex AndroidLab dataset where apps contain many redundant UI elements, the reduced upload volume offsets the extra communication overhead, resulting in a slightly lower overall token cost. Overall, our method achieves a practical trade-off among success rate, upload reduction and cost, with only a slight runtime increase and comparable cloud token usage due to the collaborative design.

**Impact of Local Model Choice.** As shown in Table 1, the performance of our method correlates with the abilities of the local LLM. Gemma2-9B yields the best results among local baselines, followed by Qwen2.5-7B and LLaMA3.1-8B. When combined with GPT-4o in our method, stronger local LLMs lead to higher success and reduction rates, with the Gemma combination achieves near GPT-4o success rate (69.23%, 37.76%) with substantial reduction (55.60%, 34.96%) on two datasets. Nonetheless, even a weak model like LLaMA, sees substantial gains over its standalone baseline on both datasets(+39.86%, +23.47%), confirming the effectiveness of collaboration. Our framework lowers the burden on local LLMs by dividing UI into blocks and avoiding full-screen reasoning. This explains why Qwen, which achieves the lowest 5.10% success rate as a local baseline on AndroidLab, can reach the highest 41.84% success rate along with a 29.97% reduction when paired with GPT-4o.

**Impact of Dataset Difficulty.** The two datasets differ significantly in difficulty, as reflected by the cloud LLM performance. The GPT-4o baseline achieves 74.13% success rate on DroidTask but only 44.90% on AndroidLab. This is due to differences in apps, task descriptions and minimum steps required. On the relatively easier DroidTask dataset, our method achieves a high success rate of

69.23% and a reduction rate of 55.60%. On the more challenging AndroidLab dataset, our 41.84% success rate remains close to GPT-4o-only performance. Although the reduction rate drops, it still remains around 30%. This shows that our method generalizes well across tasks of varying complexity.

**Performance Across different Apps.** Figure 5 presents a detailed per-app comparison of task success rate and UI reduction rate between our method and the GPT-4o baseline on both datasets. As expected, the GPT-4o baseline slightly outperforms our method on most apps (e.g., Gallery, Notes), since it has full access to UI context. Interestingly, we surpass GPT-4o on some apps (e.g., Firefox, Map), largely because we filter out irrelevant UI elements that might distract or mislead the LLM. E.g., complex location data in Map may interfere with reasoning, while our method eliminates such noise and focuses on task-relevant content. Our reduction rates clearly outperform GPT-4o (0%) but vary across apps, influenced by both task difficulty and UI structure. Simple tasks on Clock require little context, allowing us to aggressively reduce UI elements without sacrificing performance. Apps with flat layouts or unbalanced XML structures (Settings, Pimusic) affect our block partitioning and limit reduction.

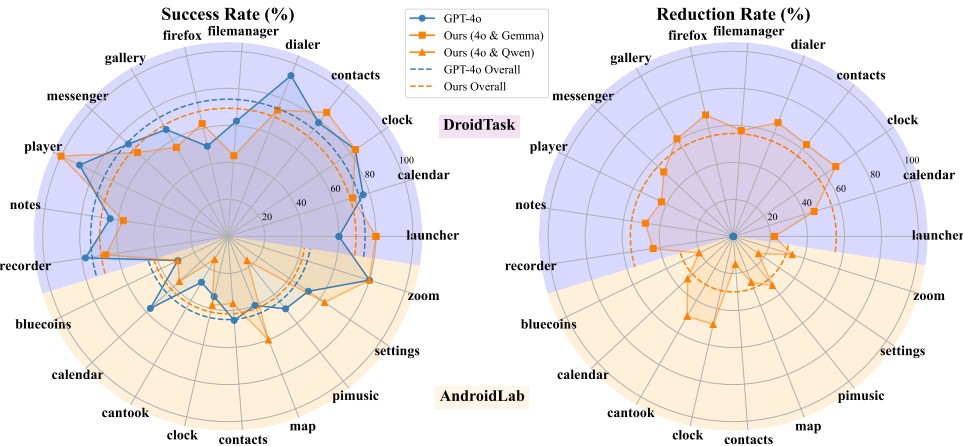

Figure 5: Success and reduction rates of the GPT-4o baseline and our method (using best-performing setting) across apps on both datasets. Note: On the right, GPT-4o appears as a single point due to 0% reduction. Dashed arcs indicate the overall performance on the entire dataset.

## 4.3 Ablation Study

Table 3: Ablation study results of different design modules over the DroidTask dataset.

|  | Success Rate | Δ | Reduction Rate | Δ |
|---|---|---|---|---|
| w/o our block partitioning | 62.24% | -6.99% | 53.44% | -2.15% |
| w/o our co-planning | 59.44% | -9.79% | 48.67% | -6.93% |
| w/o multi-round interactions | 36.36% | -32.87% | 52.57% | -3.03% |
| w/o accumulation mechanism | 53.15% | -16.08% | 52.57% | -3.03% |
| Random ranking | 51.75% | -17.48% | 35.87% | -19.73% |
| Basic-order ranking | 46.15% | -23.08% | 32.59% | -23.01% |

We conduct an ablation study on the DroidTask dataset to isolate the contribution of each module in our collaborative framework. All experiments use the "Ours (4o & Gemma)" setting as the reference.

**Block Partitioning.** Replacing our XML-layout-informed partitioning with a simple equal split of all UI elements into three groups causes the success rate to drop by 6.99% and the reduction rate by 2.15%. This result confirms that leveraging the XML layout to group semantically related elements can help our framework to do better block filtering.

**Co-planning.** We remove the co-planning module so that no guided sub-tasks are generated collaboratively and the local model must directly rank page blocks on its own. Due to limited planning ability,

the local LLM often mis-ranks important blocks on the first pass. These early errors not only reduce accuracy (-9.79%) but also trigger additional rounds of interaction in the co-decision-making stage, further harming upload reduction (-6.93%). This ablation underscores the critical role of sub-task co-planning in guiding the local LLM to better rank UI blocks.

**Co-decision-making.** Removing multi-round interactions, where the cloud LLM only receives the single top-ranked block from the local LLM without requesting further blocks, causes a 32.87% drop in success rate and a 3.03% drop in reduction rate. Similarly, removing the accumulation mechanism (where the cloud LLM, after finding a block inadequate, requests another block from the local model but makes decisions based only on the new block) results in performance decrease (-16.08% success, -3.03% reduction). These results demonstrate that our co-decision-making is essential both for mitigating the effects of mis-ranked blocks from the local LLM and for improving the cloud LLM's understanding and decision accuracy on the incomplete page context.

**Ranking strategies.** Replacing the local LLM's ranking with random ordering reduces success by 17.48% and reduction by 19.73%. Using a simple top-to-bottom left-to-right ("basic-order") block ranking performs even worse (-23.08% success, -23.01% reduction). These naive strategies often surface irrelevant blocks unrelated to the task, which not only misleads the cloud model's decisions but also forces additional requests to retrieve more useful blocks. This demonstrates that the local LLM's ranking plays a critical role in efficiently filtering informative content.

# 5   Related Work

Existing LLM-based GUI operation agents [17, 44, 34, 38, 18] for task automation span Mobile [31, 36, 33, 32, 46, 14, 2, 1, 15, 37, 13, 35], Web [6, 49, 8, 10, 42], and Desktop [45, 22, 39, 29] platforms. We focus on Mobile agents. Prior work can be categorized along two axes: (1) GUI representation, including screenshot-based (via Multimodal LLM) [33, 32, 3, 48, 21, 11, 43, 16, 27, 24] and structured XML-based [31, 36, 37, 14] methods; (2) LLM type, including approaches using closed cloud-based models [36, 33, 32, 46, 14] and smaller open local-deployable models [3, 48, 21, 37]. Autodroid [36] and MobileGPT [14] adopt XML-based representations and enhance cloud LLMs by incorporating app-specific domain knowledge through prompting. MobileAgent [33] leverages screenshots and cloud-based multimodal LLMs (MLLMs) for reasoning, with its v2 [32] introducing a multi-agent architecture to enhance performance. Some local methods [3, 48, 21, 4] focus on improving local LLM's grounding ability to locate corresponding UI elements given user instructions by finetuning on GUI datasets [28, 26, 19]. However, even with such tuning, local models often struggle to match the performance of closed cloud models or fail to generalize well, and some of them require substantial GPU resources. Despite these limitations, they completely avoid exposing UI content to the cloud. Our method targets a balance between accuracy and UI exposure via collaboration between cloud and local LLMs. It is XML-based, training-free, and can be integrated into existing works.

# 6   Conclusion and Future Work

Current cloud-LLM agents achieve high task success rates but expose excessive UI information. Local-LLM agents avoid cloud exposure but suffer from limited performance and generalization. We propose CORE, a collaborative XML-based framework that balances accuracy and UI exposure by leveraging both local and cloud LLMs, achieving GPT-4o-level accuracy with much less UI exposure. Looking forward, the rapid development of multimodal GUI agents opens opportunities for extending CORE beyond XML to vision-based pipelines. We present initial experiments in the Appendix H, and future work will explore pure-visual pipelines and broader multimodal integrations.

## Acknowledgments and Disclosure of Funding

This work was supported in part by National Key R&D Program of China (No. 2022ZD0119100), China NSF grant No. 62025204, No. 62202296, No. 62272293, No. 62441236, No. U24A20326, and No. 62572299, Alibaba Innovative Research (AIR) Program, and SJTU-Huawei Explore X Gift Fund. The opinions, findings, conclusions, and recommendations expressed in this paper are those of the authors and do not necessarily reflect the views of the funding agencies or the government.

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

## Appendix Contents

# A  Pseudo Code of Block Partitioning

Algorithm 2 illustrates the block partitioning process and outputs a grouping map $G$, where each key corresponds to an ancestor node ID in the XML tree, and the associated value is a list of UI element indices that can be grouped together. Using these indices, we partition all the UI elements into $k = |G|$ UI blocks, denoted as $B = [b_1, b_2, ..., b_k]$.

---

**Algorithm 2:** Layout-aware Block Partitioning

---

**Input:** Root element $e$ of an XML tree
**Output:** Grouping map $G$ from ancestor node ID to a list of important nodes' IDs

**Function** `extract_ancestor_paths(e, A, M)`:
  Append $e.index$ to $A$;
  **if** *e is interactable and semantically meaningful* **then**
    | $M[e.index] \leftarrow A[:-1]$;
  **foreach** *child c in $e.children$* **do**
    | `extract_ancestor_paths(c, A, M)`;
  Remove the last element from $A$;

**Function** `group(M)`:
  $L \leftarrow$ maximum length of any value in $M$;
  **for** $i \leftarrow 0$ **to** $L - 1$ **do**
    $G \leftarrow \{\}$;
    **foreach** $(node, path)$ *in* $M$ **do**
      **if** $i < |path|$ **then**
        | $g \leftarrow path[i]$;
      **else**
        | $g \leftarrow path[-1]$;
      Append $node$ to $G[g]$;
    **if** $|G| \geq 3$ **then**
      | **return** $G$
  **return** $G$

**Main:**
$A \leftarrow [\,]$;                                    // List to store current ancestor path
$M \leftarrow \{\}$;                       // Map from important node to its ancestor path
`extract_ancestor_paths(e, A, M)`;
$G \leftarrow$ `group(M)`;
**return** $G$

---

# B  More Implementation Details

## B.1  Details of Local LLMs

To ensure manageable memory usage and efficiency, we use the following quantized GGUF versions: Gemma2-9B-Instruct-Q5_K_M, Qwen2.5-7B-Instruct-Q6_K_L, and LLaMA3.1-8B-Instruct-Q6_K, each reduced to approximately 6.5GB on disk.

## B.2  Details of Action Space

In our work, the agent supports the following action types: Click, Long Click, and Input Text. More complex interaction types are not considered. To handle scrollable pages, we adopt a specific strategy: if the cloud LLM is unable to make a decision for the current step after the co-decision-making stage of our framework, we attempt to scroll the page. The framework then processes the new UI state, repeating this process until no further scrolling is possible. To prevent infinite scrolling, we set an upper limit on the number of scroll attempts per round.

## B.3  Agent Prompt

The prompts used by the local and cloud LLMs in our framework for collaborative planning and decision-making are presented below.

**Prompts used for collaborative planning:**

---

**Prompt for generating the sub-task candidate of each UI block (local LLM):**

You are a Planner, skilled at analyzing mobile UI states and task progress. Given a task description, previous UI actions, and part of current UI state, your job is to provide the most appropriate current step instruction.

You receive the task description from the user: [Task].
The previously completed steps for the task include: [History].
This is one section of current UI state: [UI Block State].

You must give the current step instruction based on the given section of current UI state(others are masked for privacy), even if the section seems irrelevant. Your response should be a single, precise current step instruction, it can't involve precise UI element information but should involve a logic task explanation, focusing only on what needs to be done immediately in the current UI state to progress the task.

---

**Prompt for confirming the best sub-task (cloud LLM):**

You are a Planner, skilled at analyzing mobile UI states and task progress. Based on a whole task description and previous UI actions, you need to give the current step instruction focusing only on what needs to be done immediately. A weaker local LLM has generated several current step instructions based on different sections of current UI state. Due to its weaker ability and incomplete information, some of them may be wrong. You can't see any private UI state, but the weaker local LLM can. You can analyze based on its generated current step instructions.

You receive the task description from the user: [Task].
The previously completed steps for the task include: [History].
The weaker local LLM generated several current step instructions based on different part of current UI: [Sub-task Candidates].

You can choose a most appropriate one from them, if you think all of them are wrong, you can correct them and give a new correct current step instruction. Never output other explanations. If you think the task has been finished, output FINISHED only.

---

**Prompts used for collaborative decision-making:**

---

**Prompt for ranking UI blocks (local LLM):**

You are a smartphone assistant to help users complete tasks by interacting with mobile apps. The current UI state is shown below. It is separated into several sections, and each section is composed by several UI elements.

Current UI state: [UI State]
Given a current task, and the sections of part of current UI state, your job is to score each section to judge the probability that it can solve or progress current task: [Sub-task].

In most time, elements grouped in one section are relevant. Sometimes, maybe only one UI element is most useful and other elements in the same section are irrelevant, this section should still be assigned high score. Output json like {"0": "<score of section 0>", "1": "<score of section 1>", ...}, the score should be a float between 0 and 1 and sum up to 1. Also output your explanation briefly.

---

## C  Details of Datasets

Table 4 and Table 5 show detailed task descriptions of DroidTask [36] and AndroidLab [40] datasets. For DroidTask, we select 143 tasks from 12 compatible apps, with some task instructions revised for improved clarity. For AndroidLab, we include 98 operation tasks spanning 9 apps.

Table 4: 143 tasks across 12 apps selected from DroidTask dataset.

| App | Task |
|---|---|
| Applauncher | Search 'Clock' app and open it. |
| | Sort apps by title in descending order. |
| | Change column count to 3. |
| | Change theme color to light. |
| | Disable closing this app when launching a different one. |
| Calendar | Create a new event, the task is 'laundry', save it. |
| | Use 24 hour formats in the Calendar app. |
| | Export the events to test.ics under DCIM folder. |
| | Change the event type 'regular event' to 'deadlines'. |
| | Change the event reminder sound of Calendar app to 'Bell'. |
| | Disable start week on Sunday. |
| | Add the event of 'VisitParents' on any day, remind me 1 hour before, save it. |
| | Create an event of 'homework' with daily repetition and save it. |
| | Search the task 'homework'. |
| | Delete all the events in the Calendar app. |
| | Change the view from monthly to yearly. |
| | Add the holidays of China to the calendar. |
| | Add the holidays of South Africa to the calendar and list all the events. |
| | Add all my contacts' birthdays into the calendar. |
| | Add the holidays of United States to the calendar. |
| | Change the snooze time to 30 min. |
| | Add all the contacts anniversaries into Calendar. |
| Clock | Change alarm max reminder duration to 1 minute. |

| | |
|---|---|
| | Disable 'start week on sunday'.
Change timer max reminder duration to 5 minutes.
Turn off increase volume gradually.
Turn off always use same snooze time.
Do not prevent the phone from entering sleep mode when the application is running in the foreground.
Turn on the '7:00 am' alarm clock vibration function.
Open the Stopwatch page, then go back to Clock page.
Set the sorting order for alarms by 'Day and Alarm time'.
Set the snooze time to 5 minute.
Add a new timer of 5:00.
Check the frequently asked questions. |
| Contacts | Create a new contact Stephen Harry, mobile number 12345678900.
Change the font size of the Contacts app to medium.
Sort contacts based on 'Date created' in descending order.
Change Stephen Harry's mobile number from '12345678900' to '222222'.
Delete contact Stephen Harry.
Change settings to show phone numbers in the list view.
Open dialpad and call 123.
Change theme color to light.
Disable showing the dial pad button on the main screen.
Open Bob's information page and call him.
Open Bob's information page, then send a text message to him with the following content: 'Good morning' (Don't retry if not sent).
Open Bob's information page, then add him to Favorites.
Export contacts to a .vcf file named 'classmate'.
Create a new group 'Classmates', add Alice, Bob and Jack to the new group, and text to the group 'GatheringInTheOldPlace' (Don't retry if not sent). |
| Dialer | Create a new contact John Smith, mobile number 123456789.
Call 123.
Change settings to turn on hide dialpad numbers.
Text Alice that 'I love you'. (Don't retry if not sent)
Delete all call history.
Modify Jack's mobile number to 654321 and save it.
Switch custom colors to light and save it.
Search Alice first, then check her info.
Open Alice's information page, then add her to favorites.
Delete contact Jack.
Sort contacts by first name in descending order.
Call Alice.
View favorite contacts.
Switch to call history page, then search Bob.
Adjust font size to medium. |
| FileManager | Check the total storage of my phone.
Go to the 'Recents' tab and open arbitrary file.
Disable showing the 'recents' tabs
Change font size of the File manager app to large.
Change pressing 'back' from twice to once to leave the app.
Go to the 'Files' tab and open arbitrary folder.
Go to settings to add 'Alarms' folder as favorite.
Go to 'Download' folder and open Diary.pdf.
Open calendar.ics in 'Download' folder with calendar, open with calendar just once.
Sort the folders by size.
Sort the folders by last modified in descending order.
Change the storage type to sd card.
Open 'Android' folder, sort the files by extension.
Change the view type to grid. |

| | |
|---|---|
| | Temporarily show hidden files.
Change the theme to light. |
| Firefox | View files downloaded through firefox.
Set firefox as the default browser.
Open dark theme.
Check add-ons.
Check bookmarks on firefox.
Check firefox version
Go to the browsing history page.
Change language to Simplified Han. |
| Gallery | Open DCIM folder in gallery app.
Sort folders in the Gallery app by name in ascending order.
Create a folder and name it 'traveling_photos' in the 'DCIM' folder, the folder path is 'Internal/DCIM/'.
Search for the DCIM folder in gallery app first, then open it.
Do not display GIF images in the Gallery app.
Change the display mode from grid to list in the Gallery app.
Display five images per row in the Gallery app.
Show all contents in all folders within the gallery app.
Restore the video to its previous playback position the next time it is opened. |
| Messenger | Find messages with Alice.
Remove accents and diacritics at sending messages.
Send message by pressing Enter.
Enable delivery reports.
View messages with Bob.
Send long messages as MMS.
Switch custom colors to light and save it.
Send a text message 'Morning' to Alice (Don't retry if not sent).
Export all messages.
Mark the chat with Bob as Unread.
Call Alice.
Change the name of the chat with Alice to 'team_discussion'.
Show a character counter at writing messages.
Delete Bob's message records.
Adjust font size to large. |
| MusicPlayer | Change the theme of the Music player app to light and save it.
Search for the song 'Last Stop' and play it.
Set the sleep timer for five minutes.
Play the song 'Last Stop' in the tracks.
Sort by year in descending order in Albums page.
Play 'Last Stop' in playlist 'All tracks' and set the loop mode to 'loop the current song'.
Change the mode to 'Hip Hop'.
Set the song to gapless playback.
Check the album 'In The Woods Vol.1' details in the Albums. |
| Notes | Create a text note named 'NewTextNote'.
Print 'NewTextNote' as NewTextNote.pdf
Create a text note called 'test', type '12345678', and search for '234'.
Export note 'test' as file test.txt, only export the current file content.
Delete the note 'test'.
Show the number of words in the Notes app.
Create a checklist named 'test1' and sort the items by creating date.
Change the alignment to center.
Set app theme to light and save it.
Create a new note called 'test2' and type '123456'.
Rename the note 'test2' to 'events'.
Adjust the fontsize of the Notes app to 125%.
Disable autosave notes |

| | Create a checklist note called 'NewCheckList'. |
|---|---|
| Voicerecorder | Hide the recording notification. |
| | Search for 'test1' and click on the 'test1.m4a'. |
| | Change the theme color to white and save it. |
| | Open 'frequently asked questions'. |
| | Set extension to mp3. |
| | Set bitrate to 96 kbps. |
| | Start recording automatically after launching the app. |
| | Use player to play 'test1.m4a' |
| | Set audio source to microphone. |

Table 5: 98 tasks across 9 apps selected from AndroidLab dataset.

| App | Task |
|---|---|
| Bluecoins | Log an expenditure of 512 CNY in the books. |
| | Record an income of 8000 CNY in the books, and mark it as 'salary'. |
| | Note down an expense of 768 CNY for May 11, 2024. |
| | For March 8, 2024, jot down an income of 3.14 CNY with 'Weixin red packet' as the note. |
| | For May 14, 2024, record an expenditure of 256 CNY, marked as 'eating'. |
| | Adjust the expenditure on May 15, 2024, to 500 CNY. |
| | Shift the income entry from May 12th, 2024, to May 10th, 2024, and update the amount to 18,250 CNY. |
| | Switch the May 13, 2024, transaction from 'expense' to 'income' and add 'Gif' as the note. |
| | Change the type of the transaction on May 2, 2024, from 'income' to 'expense', adjust the amount to 520 CNY, and change the note to 'Wrong Operation'. |
| | Move the expense entry from May 12, 2024, to May 13, 2024, adjust the amount to 936.02 CNY, and update the note to 'Grocery Shopping'. |
| Calendar | I want to add an event at 5:00PM today, whose Title is 'work'. |
| | Arrange an event titled 'homework' for me at May 21st, and set the notification time to be 10 minutes before. |
| | Help me arrange an event titled 'meeting' at May 13th with note 'conference room B202'. |
| | Arrange an event which starts at 2024/6/1 and repeats monthly. |
| | Edit the event with title 'work', change the end time to be 7:00 PM. |
| | Add the note 'classroom 101' to the event 'homework'. |
| | Change the notification time of event 'meeting' to be 5 minutes before and 10 minutes before. |
| | Edit the event titled 'work' and add a Note 'computer' to it. |
| | For the event titled 'work', please help me set recurrence to be daily. |
| | Arrange an event 'this day'. |
| | Edit the event titled 'this day', and make it repeat weekly. |
| | Help me add a note 'Hello' to the event titled 'this day'. |
| | Arrange an event titled 'exam'. |
| | Edit the event titled 'exam' and make it an all-day event. |
| Cantook | Import Alice's Adventures in Wonderland from folder /Download/Ebooks/. |
| | Delete Don Quixote from my books. |
| | Mark Hamlet as read. |
| | Mark the second book I recently read as unread. |
| | Open Romeo and Juliet. |
| | Open the category named 'Tragedies'. |
| | Create a new collection called 'Favorite'. |
| Clock | Set an alarm for 3PM with the label 'meeting' using Clock. |
| | Set an alarm for 6:45AM, disable vibrate and change ring song to Argon. |
| | Help me set an alarm every Monday to Friday, 7AM in morning. |

| | |
|---|---|
| | Change my clock at 9AM, make it ring everyday.
Help me set an alarm at 10:30AM tomorrow.
I need to set a 10:30PM clock every weekend, and label it as 'Watch Football Games' to remind me.
Turn off all alarms.
Delete all alarms after 2PM.
Turn off the alarm at 4PM.
Add London and Barcelona time in clock.
Delete Barcelona time from clock.
Set a countdown timer for 1 hour 15 minutes but do not start it.
Set bedtime for 10PM to sleep, wake up at 7AM.
Set sleep sounds to deep space.
Turn on the Wake-up alarm in Bedtime.
Set alarm style to Analog.
Change home time zone to Tokyo in clock.
Modify silence after to 5 minutes.
Open clock app.
Close my 7:30AM alarm.
Set an alarm at 3PM. |
| Contacts | Add John as a contacts and set his mobile phone number to be 12345678.
Add a contacts whose first name is 'John', last name is 'Smith', mobile phone number is 12345678, and working email as 123456@gmail.com.
Add a contacts whose name is Xu, set the working phone number to be 12345678 and mobile phone number to be 87654321.
Add a contacts named Chen, whose company is Tsinghua University.
Create a new label as work, and add AAA, ABC into it.
Add a work phone number 00112233 to contacts ABC.
Add birthday to AAA as 1996/10/24.
Set contacts ABC's website to be abc.github.com.
Edit a message to ABC, whose content is 'Nice to meet you', but do not send it.
Call ABC.
Delete contacts AAA. |
| Map | Add the address of openai to my Work place.
Navigate from my location to Stanford University.
Navigate from my location to University South.
Navigate from my location to OpenAI.
Navigate from my location to University of California, Berkeley. |
| Pimusic | Play the first song in 'Favorite' playlist.
Sort Pink Floyd's songs by duration time in descending order.
Create a playlist named 'Creepy' for me.
Pause the currently playing song and seek to 1 minute and 27 seconds.
Play Lightship by Sonny Boy.
Sort the songs by duration time in ascending order. |
| Settings | Turn on airplane mode of my phone.
I do not want turn on wifi automatically, turn it off.
Set private DNS to dns.google.
Turn off my bluetooth.
Change my bluetooth device name to 'my AVD'.
Show battery percentage in status bar.
Turn my phone to Dark theme.
Change my Brightness level to 0%.
I need to close down my Ring & notification volume to 0%.
Set my alarm volume to max.
Change text-to-speech language to Chinese.
Set current time of my phone to 2024-5-1.
Turn off Ring vibration.
Add Español (Estados Unidos) as second favorite languages. |

| | Check Android Version. |
|---|---|
| | Disable Contacts' APP notifications. |
| | Check my default browser and change it to firefox. |
| | Uninstall booking app. |
| | Open settings. |
| Zoom | Join meeting 1234567890. (You should not click join button, and leave it to user) |
| | Join meeting 0987654321, and set my name as 'Alice'. (You should not click join button, and leave it to user) |
| | I need to join meeting 1234567890 without audio and video. (You should not click join button, and leave it to user) |
| | Set auto connect to audio when wifi is connected in zoom settings. |
| | Change my reaction skin to Medium-light in zoom settings. |

# D   Effect of Limiting the Number of UI Blocks on Performance

To study the trade-off between task success rate, UI exposure reduction, and computational cost, we limit the number of UI blocks to at most three. Based on our original block partitioning strategy, we merge the resulting blocks to ensure that no more than three blocks remain. We evaluate this modified method on both the DroidTask [36] and AndroidLab [40] datasets and compare it against our original framework. As shown in Table 6 and Table 7, limiting the number of blocks has minimal impact on the task success rate. As expected, the reduction in the number of blocks leads to a decrease in the UI exposure reduction rate. On the positive side, both latency and token usage decrease. This is because the maximum number of generated sub-task candidates decreases (<=3) during co-planning, and during co-decision-making, the maximum number of interactions between the local and cloud LLMs is limited to three. This demonstrates a trade-off between efficiency and UI reduction.

Table 6: Results under restriction of three blocks and comparisons to the original version.

| Methods | DroidTask Dataset | | | | | | | |
|---|---|---|---|---|---|---|---|---|
| | Success Rate | Δ | Reduction Rate | Δ | Latency(s) | Δ | Cloud Tokens | Δ |
| Ours* (GPT-4o + Gemma 2-9B) | **68.53%** | -0.7% | **41.44%** | -14.16% | 8341.47 | -9.88% | 656959 | -21.96% |
| Ours* (GPT-4o + Qwen 2.5-7B) | 62.94% | +1.4% | 37.16% | -4.73% | 7575.88 | -13.54% | 701274 | -15.42% |
| Ours* (GPT-4o + LLaMA 3.1-8B) | 49.65% | 0% | 37.33% | -3.23% | 8773.23 | -1.05% | 716769 | -16.89% |

Table 7: Results under restriction of three blocks and comparisons to the original version.

| Methods | AndroidLab Dataset | | | | | | | |
|---|---|---|---|---|---|---|---|---|
| | Success Rate | Δ | Reduction Rate | Δ | Latency(s) | Δ | Cloud Tokens | Δ |
| Ours* (GPT-4o + Gemma 2-9B) | **39.80%** | +2.04% | 26.79% | -8.17% | 6279.42 | -12.76% | 761184 | -10.59% |
| Ours* (GPT-4o + Qwen 2.5-7B) | **39.80%** | -2.04% | 25.59% | -4.38% | 5494.08 | -15.10% | 717340 | -13.28% |
| Ours* (GPT-4o + LLaMA 3.1-8B) | 30.61% | -4.08% | **27.27%** | -4.38% | 6572.95 | -8.61% | 785490 | -0.73% |

# E   Evaluation Results under Different Reduction Metrics

We provide multiple UI reduction metrics under different comparison settings to enable a comprehensive and fair assessment of our UI reduction framework. The extended results of these metrics, which complement the main results presented in the main paper, are shown in Table 8.

*Reduction Rate (RR).* This is the primary metric used in the main body of our paper. It measures the reduction in the number of UI elements uploaded to the cloud compared to the GPT-4o baseline. To ensure fairness, we consider only the rounds where both methods make the same decision on the same UI screen.

*RR-1.* This is a relaxed variant of RR. It measures the reduction in the number of UI elements uploaded to the cloud compared directly to the GPT-4o baseline, without requiring the two methods to align on the same interaction rounds or UI states. All rounds are included in the comparison.

*RR-2.* This metric evaluates the reduction in UI elements by comparing against the total number of UI elements on the full original page, without referencing the GPT-4o baseline. It only includes

pages that can be successfully partitioned into multiple blocks by our method. Pages that cannot be partitioned and consist of only a single block are excluded.

*RR-3.* This metric extends RR-2 and is more inclusive. It measures the percentage of reduction in uploaded UI elements relative to the total number of UI elements on the original page, and includes all cases — even those that cannot be partitioned and consist of a single block.

Table 8: Extended results under different UI reduction metrics.

| Methods | DroidTask Dataset | | | | AndroidLab Dataset | | | |
|---|---|---|---|---|---|---|---|---|
| | RR | RR-1 | RR-2 | RR-3 | RR | RR-1 | RR-2 | RR-3 |
| Ours (GPT-4o + Gemma 2-9B) | 55.60% | 52.02% | 55.63% | 46.24% | 34.96% | 46.28% | 35.84% | 31.09% |
| Ours (GPT-4o + Qwen 2.5-7B) | 41.89% | 47.35% | 47.67% | 37.76% | 29.97% | 45.46% | 32.88% | 27.80% |
| Ours (GPT-4o + LLaMA 3.1-8B) | 40.56% | 59.76% | 51.75% | 39.86% | 31.65% | 63.07% | 41.80% | 37.81% |

# F  Case Study

We illustrate five representative examples in Figures 6–10 (three tasks from the DroidTask dataset [36] and two from the AndroidLab dataset [40]). The agent first triggers the main activity of an app to start. A hand icon indicates user-like interactions with UI elements on the screen executed by the mobile agent, and orange dashed lines represent the sequence of interactions, continuing until our framework determines task completion during co-planning. Our framework is XML-based, which extracts attributes of UI elements (e.g., `text`, `content-desc`, `bounding box`) from XML nodes, without relying on visual screenshots. However, for better illustration of UI exposure reduction, we

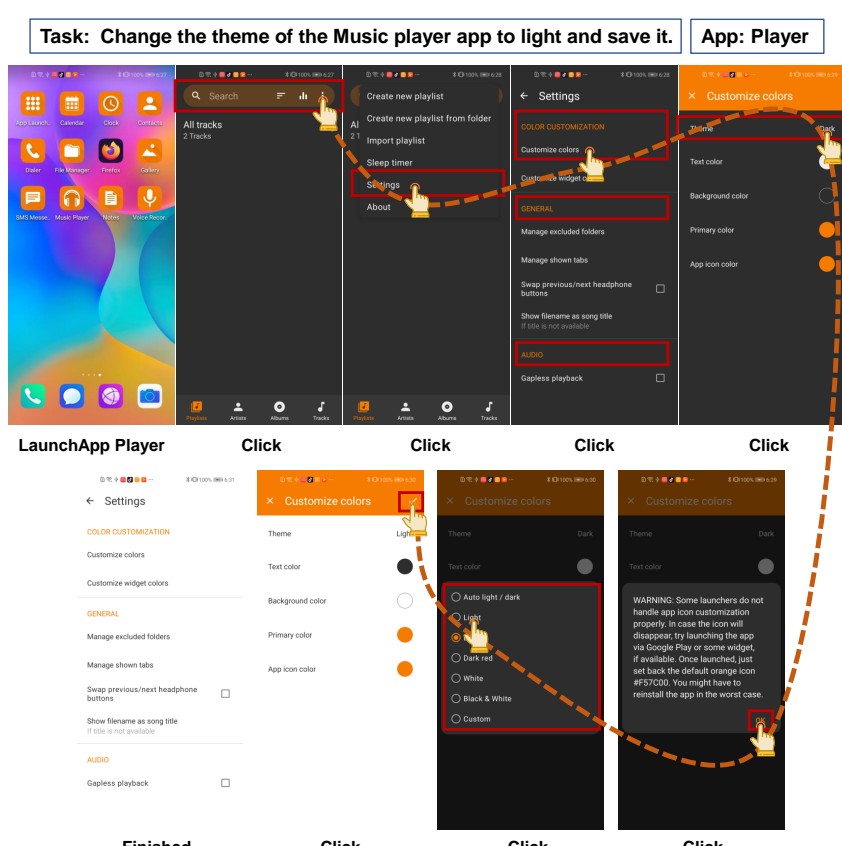

Figure 6: A case of customizing the color theme in the Player app. The task is completed with reduced UI exposure; only UI content within the red boxes is transmitted to the cloud.

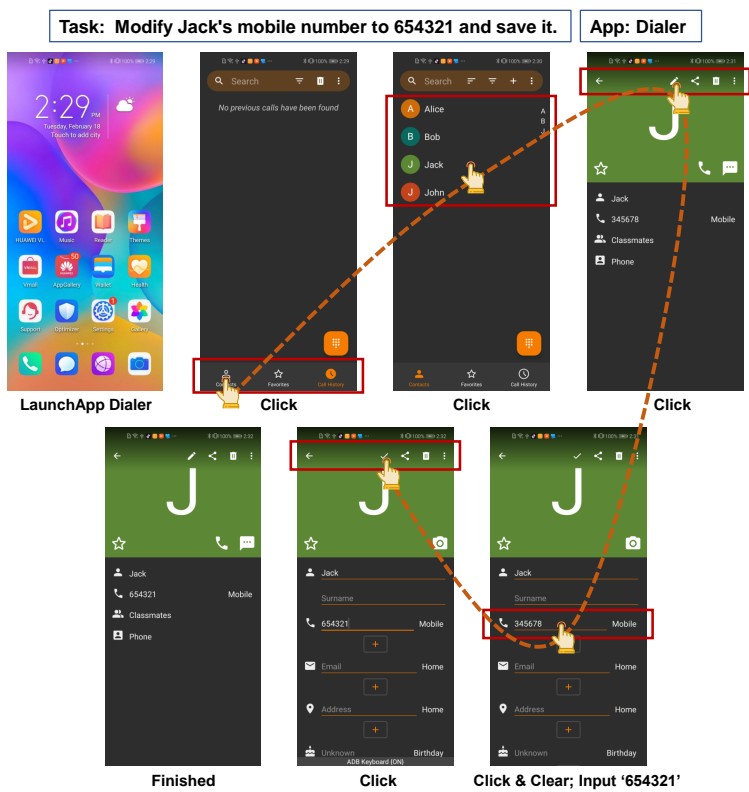

Figure 7: A case of updating a contact's phone number in the Dialer app.

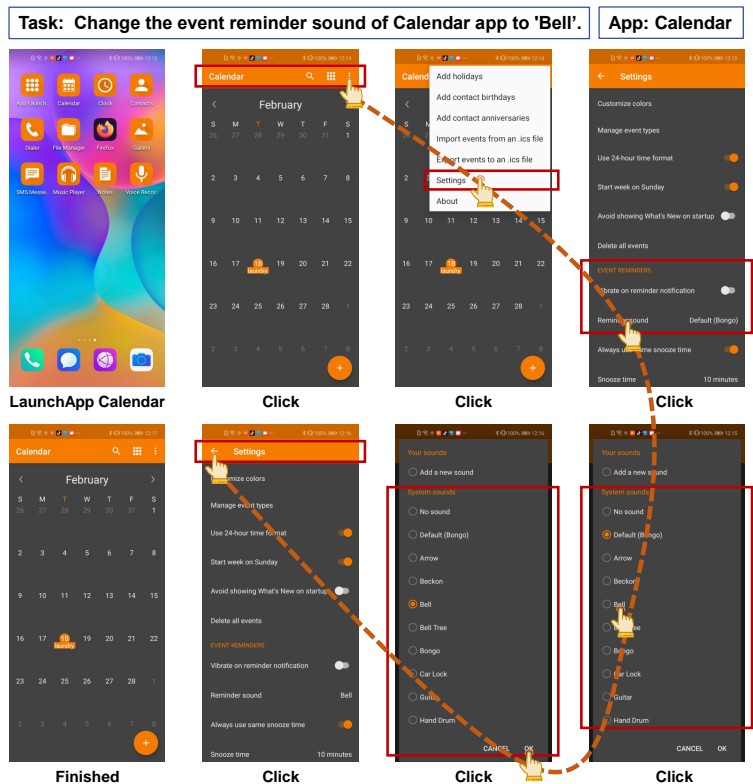

Figure 8: A case of changing the reminder sound in the Calendar app.

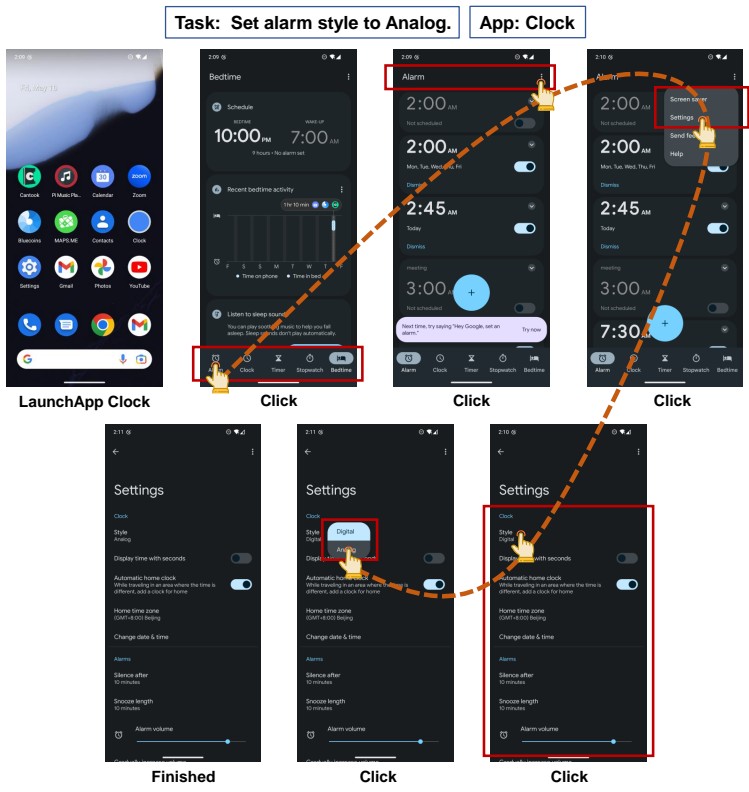

Figure 9: A case of setting the alarm type in the Clock app.

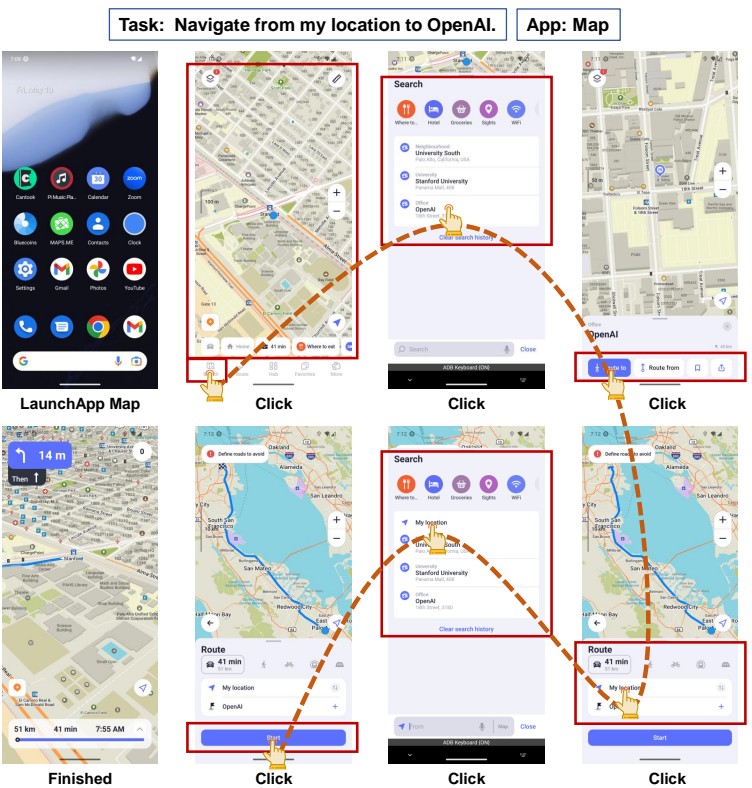

Figure 10: A case of navigation in the Map app.

visualize the subset of UI content uploaded to the cloud with red solid bounding boxes. UI elements outside these boxes are excluded from transmission to the cloud, representing the reduced exposure achieved by our method.

Overall, these tasks are successfully completed with little UI exposure. The reduction ratio in each round varies, depending on the UI layout and the number of UI blocks that the cloud LLM needs to make a confident decision on the current sub-task.

# G    On-Device LLM Deployment and Overhead Analysis

To further assess the feasibility and efficiency of local LLM inference on mobile devices, a quantized Qwen2.5-7B-Instruct model was deployed on a Xiaomi 15 Pro smartphone using Alibaba's MNN [20] mobile inference engine. All five tasks in the Applauncher app from the DroidTask dataset were selected for overhead evaluation. The hardware configurations of the smartphone, as well as the inference latency, CPU utilization, and memory usage per task, are summarized below.

Table 9: Hardware specifications of the mobile device.

| Device | DRAM | SoC | CPU |
|---|---|---|---|
| Xiaomi 15 Pro | 16 GB | Qualcomm Snapdragon 8 Elite | 2×Oryon (4.32 GHz) + 6×Oryon (3.53 GHz) |

Table 10: Performance of Qwen2.5-7B-Instruct deployed on Xiaomi 15 Pro (MNN).

| Model | Prefill Time/task (s) | Decode Time/task (s) | Memory (GB) | CPU Util. (%) |
|---|---|---|---|---|
| Qwen2.5-7B-Instruct | 75.05 | 65.99 | 6.9 | 770 |

The results indicate that the average inference latency of the on-device LLM is approximately 140s per task (multiple steps), utilizing around 8 cores and maintaining memory consumption below 7 GB.

To provide a broader comparison, the more general llama.cpp inference framework was also adopted to deploy all three local LLMs used in the evaluation, including Qwen2.5-7B-Instruct, Llama-3.1-8B-Instruct, and Gemma-2-9B-Instruct. The latency of Qwen2.5-7B-Instruct under llama.cpp was found to be roughly 5× higher than that using MNN, while the CPU and memory consumption remained comparable across the two engines. In addition, the overhead increases slightly with the model size increasing from 7B to 9B.

Efficient on-device inference remains a significant challenge in mobile agent systems, primarily due to hardware constraints. As a result, existing work typically invoked cloud LLM APIs or ran LLMs on more powerful computers and servers. Following the common practice, the main experiments were conducted on a consumer-grade RTX 4090D GPU, with the overhead results of Qwen2.5-7B-Instruct shown below. The inference speed on 4090D is 4.39× faster than using MNN on Xiaomi 15 Pro.

Table 11: Comparison of latency between baseline and local LLM deployments.

| Methods | Cloud Latency (s) | Local Latency (s) | Total Latency (s) | Ratio vs GPT-4o Baseline |
|---|---|---|---|---|
| GPT-4o (Baseline) | 40.32 | 0.00 | 40.32 | 1.00× |
| Ours (GPT-4o + Local LLM on RTX 4090D) | 29.15 | 32.12 | 61.27 | 1.52× |
| Ours (GPT-4o + Local LLM on Smartphone) | 29.15 | 141.04 | 170.19 | 4.22× |

# H    Adaptability to Screenshot-based and Multimodal Settings

Prior to 2025, foundation multimodal LLMs exhibited limited reliability in accurately localizing UI elements. Consequently, CORE was initially developed upon XML-based UI representations, which provide precise bounding box information and a structured hierarchy, both crucial for reliable decision-making in mobile UI automation.

Nonetheless, multi-modality plays a complementary and increasingly important role. XML encodes semantic metadata such as *content-description*, reflecting developer intent, whereas screenshots capture richer visual context. Recognizing this complementarity, the CORE framework was designed to

be modality-agnostic and can be seamlessly extended to incorporate screenshot-based or multimodal inputs.

The key modules of CORE, including layout-aware block partitioning, co-planning, and co-decision-making are all designed independently of the input modality. Each can operate on XML-only, screenshot-only, or hybrid representations. For instance, the block partitioning module may employ either an XML-based structural strategy or a visual segmentation approach. Similarly, the local and cloud LLMs can be replaced with multimodal LLMs without altering the collaborative workflow.

To extend CORE to multimodal settings, a straightforward masking strategy was adopted. The selective UI reduction mechanism precisely identifies which elements should be hidden from the cloud model. Screenshots of the current UI are captured, and gray masks are applied to the bounding boxes corresponding to the filtered-out elements. The resulting masked screenshot, together with textual UI element descriptions, is then provided to a multimodal cloud LLM (e.g., GPT-4o). In comparison, the GPT-4o baseline receives the full, unmasked screenshot. The multimodal extension of CORE was evaluated on DroidTask and AndroidLab datasets, using GPT-4o as the cloud LLM and Gemma 2-9B as the local LLM. Table 12 summarizes the task success rate and UI exposure reduction.

Table 12: Performance of CORE with multimodal (screenshot-based) input on DroidTask and AndroidLab datasets.

| Methods (+Screenshot) | DroidTask Dataset | | AndroidLab Dataset | |
|---|---|---|---|---|
| | Success Rate | Reduction Rate | Success Rate | Reduction Rate |
| GPT-4o Baseline | 74.13% | 0.00% | 46.94% | 0.00% |
| CORE (GPT-4o + Gemma 2-9B) | 69.93% | 56.84% | 39.80% | 37.51% |

These results indicate that the multimodal extension of CORE preserves high decision quality while substantially reducing unnecessary UI exposure, achieving reduction rates of 56.84% on DroidTask and 37.51% on AndroidLab. This confirms the generality and robustness of CORE across both XML and screenshot-based input modalities.

## I  More Evaluation Results on Other Datasets

To further verify the generality of the proposed framework, additional experiments were conducted on two challenging benchmarks: a subset of the LlamaTouch [47] dataset containing social media tasks, and the AndroidWorld [25] environment with modified XML extraction. The 64 tasks selected from LlamaTouch are detailed in Table 13, while all 116 tasks from AndroidWorld are used according to the task list provided in Appendix F of the original paper [25].

Table 13: 64 tasks across 5 apps selected from LlamaTouch dataset.

| App | Task |
|---|---|
| Instagram | Open the Instagram app and follow an account named 'artem_chek'. |
| | Open the Instagram app and view the notification list. |
| | Open the Instagram app and like the first post in the page. |
| | Open the Instagram app and navigate to your personal profile page. |
| | Open the Instagram app, navigate to your personal profile page and view your following list. |
| | Open the Instagram app, search for '#travelgoals' and follow the hashtag. |
| | Open the Instagram app and navigate to 'Settings'. |
| | Open the Instagram app, go to 'Settings' and activate 'Private Account'. |
| | Open the Instagram app and edit your profile to add 'Travel Enthusiast' to your bio. |
| | Open the Instagram app and view the message list. |
| | Open the Instagram app and follow the user of first post . |
| | Open the Instagram app, navigate to your personal profile page and view your 'followers' list. |

| | |
|---|---|
| | Open the Instagram app, navigate to your personal profile page, go to 'Settings' and view the 'Blocked' list. |
| | Open the Instagram app, navigate to your personal profile page, go to 'Settings' and view the 'Close Friends' list. |
| | Open the Instagram app, navigate to your personal profile page, go to 'Settings' and view the 'Archive' list. |
| Pinterest | Search for 'interior design ideas' on Pinterest. |
| | Open Pinterest and search a user named 'Wendy's Lookbook' and follow the user. |
| | Navigate to the settings menu in Piniterest. |
| | Create a new board named 'DIY Crafts' on Pinterest. |
| | Open Pinterest and copy the link of a pin from the 'Home' page. |
| | Open Pinterest and save a pin from the 'Home' page to your Profile. |
| | Open Pinterest and view the comments of a pin from the 'Home' page. |
| | Open Pinterest and view your own pin list. |
| | Open Pinterest and open your own profile page. |
| | Open Pinterest, open your own profile page and copy the shared link of yourself. |
| | Open Pinterest, open your own profile page and view your following list. |
| | Open Pinterst and refresh the 'Home' page to see new pins by clicking the 'Home' button again. |
| | Open Pinterest and open one of your own board. |
| | View 'Updates' in the 'Notifications' page in Pinterest. |
| | View 'Messages' in the 'Notifications' page in Pinterest. |
| | Filter 'Updates' notifications by 'Comments' in the 'Notifications' page in Pinterest. |
| | Open Pinterest and set your profile visibility to 'Private profile' in the settings menu. |
| | Open Pinterest and search 'DIY' in your own board list and open it. |
| | Open the Pinterest app, navigate to the 'Saved' page and sort your boards by 'A to Z'. |
| | Open the Pinterest app, navigate to the 'Saved' page and view the 'Privacy and data' in the settings menu. |
| | Open the Pinterest app, navigate to the 'Saved' page and view the 'Home feed tuner' in the settings menu. |
| | Navigate to the 'Saved' page in the Pinterst app and view the 'Social permissions' in the settings menu. |
| | Navigate to the 'Saved' page in the Pinterst app, open the settings menu and enable the 'Filter comments on my Pins' button in the 'Social permissions'. |
| | Open the Pinterest app, navigate to the 'Saved' page and view the 'Account management' in the settings menu. |
| Reddit | Subscribe to the subreddit r/science and set community alerts to frequent on Reddit. |
| | Find and display the hottest post from r/worldnews on Reddit. |
| | Search for all posts related to SpaceX launches last month in the r/space subreddit on Reddit. |
| | Open Reddit, search for the 'technology' subreddit, and access it. |
| | Open Reddit, search for the 'technology' subreddit and aceess to Communities. |
| | Open Reddit, search for the 'technology' subreddit and aceess to Media. |
| | Open Reddit, navigate to your Home feed and select Latest. |
| | Open Reddit, navigate to the Communities. |
| | Open Reddit, navigate to the Communities, search for openai and join. |
| | Open Reddit, navigate to create a community. |
| | Open Reddit, navigate to Inbox. |
| | Open Reddit, navigate to Notifications setting. |
| | Open Reddit, navigate to Chat. |
| | Open Reddit, navigate to Chat and explore channels. |

| | Open Reddit, check my profile. |
|---|---|
| | Open Reddit, go to setting. |
| | Open Reddit, go to history. |
| X(Twitter) | Open the X app and check my notifications |
| | Find the latest post of Openai on X. |
| | Follow Yann LeCun on X. |
| | Check my inbox on X. |
| | Check latest trendings in X app. |
| | Upload my avatar on X app using my latest picture on device. |
| | Bookmark latest post from OpenAI on X app. |
| | Check posts from my bookmark on X app. |

**Evaluation on Social Media Applications.** Social media apps typically feature highly dynamic and frequently updated UIs, posing additional challenges for layout-aware partitioning and selective UI reduction. Following this observation, experiments were performed on 64 tasks from popular social platforms, including Instagram, X (Twitter), Reddit, and Pinterest, sampled from the LlamaTouch dataset. The evaluation results are summarized in Table 14.

Table 14: Performance comparison on a subset of the LlamaTouch dataset (social media apps). The proposed framework maintains comparable success while substantially reducing UI exposure.

| Method | Success Rate | Reduction Rate |
|---|---|---|
| GPT-4o (Baseline) | 70.31% (45/64) | 0.00% |
| Ours (GPT-4o + Gemma-2-9B-Instruct) | 60.94% (39/64) | 48.94% |

The results indicate that the proposed CORE framework completed only six fewer tasks than the full-UI GPT-4o baseline, while achieving a 48.94% reduction in UI exposure. These findings confirm the effectiveness of CORE on dynamic and visually complex social media UIs.

**Evaluation on AndroidWorld.** The AndroidWorld environment was adapted by modifying its XML extraction pipeline to match the methodology described in the main text, where the system relies on preprocessing XML trees extracted via `uiautomator2`. Evaluation results are summarized in Table 15.

Table 15: Performance on the AndroidWorld environment using the modified XML extraction pipeline.

| Method | Success Rate | Reduction Rate |
|---|---|---|
| Qwen2.5-Max (Baseline) | 35.34% (41/116) | 0.00% |
| Ours (Qwen2.5-Max + Gemma-2-9B-Instruct) | 27.59% (32/116) | 36.96% |

The results show that the CORE framework achieves a 36.96% reduction in UI exposure while maintaining a reasonable task success rate (27.59%), compared to 35.34% for the full-UI Qwen2.5-Max baseline. This further demonstrates the applicability of CORE to different task environments.

