# OpenReview forum: "CORE: Reducing UI Exposure in Mobile Agents via Collaboration Between Cloud and Local LLMs"
_NeurIPS.cc/2025/Conference — NeurIPS 2025 poster_

### Official Review · Reviewer_meZ1 · 2025-06-25

**Clarity:** 4
**Significance:** 3
**Originality:** 4
**Rating:** 4
**Confidence:** 4

**Summary:**

This paper proposes CORE, a collaborative framework that combines cloud and local LLMs to reduce UI exposure in mobile agents while maintaining task accuracy. The approach leverages layout-aware block partitioning, co-planning, and co-decision-making to selectively upload only task-relevant UI elements, achieving significant exposure reduction (up to 55.6%) with minimal impact on success rates. Experimental results on public datasets demonstrate the effectiveness of the method compared to both cloud-only and local-only baselines. The work is well-motivated, technically sound, and addresses an important privacy concern in mobile agent systems.

**Questions:**

- I suggest that the authors keep the local LLM consistent in Figure 5 (right).

**Ethical Concerns:**

["NO or VERY MINOR ethics concerns only"]

**Final Justification:**

I believe all my concerns have been addressed. In addition, unlike most current GUI Agent papers, this paper is clearly written and focuses on a well defined problem, privacy protection, and includes corresponding ablation experiments. I think it should be accepted. However, the main drawback is that it is overly engineering oriented and does not generalize to image based scenarios. Therefore, I have given it a score of 4.

**Limitations:**

yes

**Quality:**

3

**Strengths And Weaknesses:**

### Strengths
- This paper addresses an important problem: how to enable effective collaboration between local LLMs and cloud LLMs while minimizing privacy exposure.
- The proposed processing pipeline is reasonable and effectively tackles the identified challenge.
- The experimental results support the authors’ claims.


### Weaknesses
- The main issue with this paper lies in its experimental setting, which differs from most mainstream approaches. The paper primarily relies on XML-based input, whereas many modern mobile applications no longer provide XML. As a result, image-based processing would be more meaningful. Approaches involving visual-based segmentation or ranking may be more practical.
-  In addition, there is a lack of detail. I recommend the authors include a more comprehensive case study, as it is otherwise difficult to understand the exact form of the sub-tasks.
- The evaluation metrics in Table 1 are limited. The authors should not only consider the usage cost of the cloud LLM, but also account for the overall system cost (e.g., local latency), as shown in Figure 4 (Left). Therefore, I suggest the authors consider optimizing the “Rank blocks by current sub-task” step using a dedicated ranking model.

---

> ### Author Rebuttal · Authors · 2025-07-31
>
> We appreciate your insightful and valuable comments. We provide reponses to all your concerns as follows.
>
> > Q1: Adaptability to image-based setting.
>
> A1: We first clarify that we initially chose to build CORE on XML-based UI representations because, **prior to 2025**, foundation multimodal LLMs lacked sufficient reliability for accurate UI element localization. In constrat, **XML provided precise bounding box information** and a structured hierarchy that is essential for accurate decision-making. We would also like to clarify that, **in practice, XML extraction works on most pages across mainstream apps**.
>
> Of course, we agree with the reviewer that visual-based agents are rapidly advancing, and we believe **XML and screenshots each have complementary advantages**. Screenshots offer richer visual context, while XML encodes semantic metadata (e.g., content-description), reflecting developer intent.
>
> In fact, **our CORE framework is modality-agnostic** and can be **conveniently adapted to incorporate screenshots**. This is because our key contribution lies in proposing a general workflow designed to address a new problem: reducing unnecessary UI exposure to the cloud LLM while preserving decision quality. All the key modules of **layout-aware block partitioning, co-planning, and co-decision-making** are **all modality-agnostic**. **Each module can operate on XML-only, screenshot-only, or combined inputs**. For example, the block partitioning module can use either an XML-based strategy or a visual segmentation approach. Similarly, the local and cloud LLMs can be replaced with multimodal LLMs. The rest of the collaborative pipeline remains unchanged. This resonates with the reviewer's suggestion regarding visual-based segmentation or ranking, but it may need some engineering efforts.
>
> To extend CORE to multimodal input, we have adopted a straightforward masking strategy: CORE’s selective UI reduction enables us to precisely identify which UI elements should be masked. We **capture screenshots of the UI and apply gray masks to the bounding boxes of elements that CORE removes, generating a masked image. This masked screenshot is then passed to a cloud multimodal LLM (e.g., GPT-4o in our evaluation)** for decision-making, alongside the textual UI element descriptions used in the original XML-based setting. The GPT-4o baseline receives the full screenshot, while our CORE uses the masked one.
>
> We show the evaluation results on DroidTask and AndroidLab as follows.
>
> **Results on DroidTask:**
> |Methods (+screenshot)|Success Rate|Reduction Rate|Cloud Latency per Task (s)|Local Latency per Task (s) | Total Latency per Task (s) |
> | :------: | :------: | :------: |:------: |:------: |:------: |
> |GPT-4o (Baseline)| 74.13% | 0.00% | 42.19 | 0.00 | 42.19 |
> |Ours (GPT-4o + Gemma-2-9B-Instruct)| 69.93% | 56.84% | 33.81 | 30.98 | 64.79 |
>
> **Results on  AndroidLab:**
> |Methods (+screenshot)|Success Rate|Reduction Rate|Cloud Latency per Task (s)|Local Latency per Task (s) | Total Latency per Task (s) |
> | :------: | :------: | :------: |:------: |:------: |:------: |
> |GPT-4o (Baseline)| 46.94% | 0.00% | 46.33 | 0.00 | 46.33 |
> |Ours (GPT-4o + Gemma-2-9B-Instruct)| 39.80% | 37.51% | 38.75 | 38.44 | 77.19 |
>
> These results show that **CORE remains effective with multimodal input**: it achieves success rates close to the GPT-4o baseline while reducing UI exposure by **56.84%** and **37.51%** on DroidTask and AndroidLab, respectively.
>
> ---
>
> > Q2: Case study to understand the exact form of the sub-tasks.
>
> A2: We **have included several practical case studies in Appendix E (Figures 1–5) of the supplementary material**, illustrating multi-step, cross-screen trajectories under our CORE. We agree with the reviewer that adding a **dedicated case study section in the main text** would greatly improve clarity. In the revision, we will include a step-by-step case study that illustrates the **sub-task generation and selection process** within the CORE framework. For example, for the trace of task *“Change theme color to light”*, the sub-tasks generated at each co-planning step are:
>
> * “Look for the settings or theme selection option.”
> * “Open Settings”
> * “Navigate to the theme color setting.”
> * “Select the light theme option.”
> * “Save the change.”
>
> These sub-tasks are generated by the **cloud LLM** (without direct UI access) using only the overall task goal, interaction history, and candidate sub-tasks proposed by the **local LLM** from different UI regions.
>
> ---
>
> > Q3: Overall system cost analysis.
>
> A3: We have **presented the cost in Section 4.2 (Cost Analysis) and Figure 4 (Left) of the submission, which includes a full breakdown of both cloud and local latency.** To provide a clearer view, we present it in the following tabular format, as shown below:
>
> **System Cost on DroidTask:**
> |Methods| Cloud Latency per Task (s) | Local Latency per Task (s) | Total Latency per Task (s) | Ratio vs GPT-4o |
> | :------: | :------: | :------: | :------: | :------: |
> | GPT-4o (Baseline) | 40.32              | 0.00               | 40.32              | 1.00x            |
> | Ours (GPT-4o + Gemma-2-9B-Instruct)          | 32.60              | 32.13              | 64.73              | 1.61x            |
> | Ours (GPT-4o + Qwen-2.5-7B-Instruct)         | 29.15              | 32.12              | 61.27              | 1.52x            |
> | Ours (GPT-4o + LLaMA-3.1-8B-Instruct)        | 23.81              | 38.19              | 62.00              | 1.54x            |
>
> **System Cost on AndroidLab:**
> |Methods| Cloud Latency per Task (s) | Local Latency per Task (s) | Total Latency per Task (s) | Ratio vs GPT-4o |
> | :------: | :------: | :------: | :------: | :------: |
> | GPT-4o (Baseline)      | 44.24              | 0.00               | 44.24              | 1.00x            |
> | Ours (GPT-4o + Gemma-2-9B-Instruct)         | 36.20              | 37.25              | 73.45              | 1.66x            |
> | Ours (GPT-4o + Qwen-2.5-7B-Instruct)         | 33.82              | 32.21              | 66.03              | 1.49x            |
> | Ours (GPT-4o + LLaMA-3.1-8B-Instruct)        | 24.70              | 48.70              | 73.39              | 1.66x            |
>
> Regarding the reviewer’s valuable suggestion to **optimize the “Rank blocks by current sub-task” step using a dedicated ranking model**, we agree this is a promising direction. However, our preliminary attempts with either using smaller general-purpose LLMs (3B) or ranking UI blocks based on semantic similarity to the current sub-task did not yield satisfactory ranking performance. We consider that **a dedicated lightweight ranking model would require task-specific training**, which should be an important direction for future work.
>
> ---
>
> > Q4: Keep the local LLM consistent in Figure 5 (right).
>
> A4: We apologize for the confusion. In Figure 5, we used different local LLMs because we prioritized the best-performing setup for each dataset in terms of task success rate. The optimal local LLM varies on DroidTask and AndroidLab datasets. In the revision, we will either keep the local LLM across both datasets consistent in the figure or clearly explain in the figure caption why different local LLMs were chosen.

---

> > ### Comment · Reviewer_meZ1 · 2025-08-06
> >
> > Thank you for the authors' response. My concerns have been addressed. I believe the framework proposed in this paper is novel and should be accepted. Good luck!

---

> > > ### Author Response · Authors · 2025-08-06
> > >
> > > Thank you sincerely for your encouraging feedback and strong support! Your thoughtful suggestions have greatly strengthened this work. We deeply appreciate your recognition of the novelty of our proposed framework and your kind recommendation for acceptance.

---

### Official Review · Reviewer_PSyJ · 2025-06-30

**Clarity:** 3
**Significance:** 2
**Originality:** 3
**Rating:** 4
**Confidence:** 5

**Summary:**

This paper proposes CORE, a collaborative framework combining local and cloud large language models (LLMs) to automate mobile tasks while limiting unnecessary UI data exposure to the cloud. CORE partitions the UI into meaningful blocks, uses the local model to filter and rank them, and lets the cloud model make final decisions within top-ranked blocks. Experiments show CORE cuts UI exposure by over 50% while keeping high task success rates.

**Questions:**

1. I have concerns about the method for selecting necessary UI elements to upload to the cloud, as there is no specific approach designed to ensure that the uploaded information is non-sensitive. Perhaps the authors could provide concrete quantitative or qualitative criteria to demonstrate that these UI elements do not contain private information.

2. I suggest including at least one case study in the main text to qualitatively demonstrate the effectiveness of the proposed method.

**Ethical Concerns:**

["NO or VERY MINOR ethics concerns only"]

**Final Justification:**

While the paper does not completely solve the privacy exposure issue in GUI automation, it is an insightful and exploratory work. Based on the quantitative privacy analysis provided by the authors, the proposed method demonstrates a meaningful reduction in privacy exposure. Considering the feedback from other reviewers, I have decided to raise my score to 4.

**Limitations:**

Yes

**Quality:**

3

**Strengths And Weaknesses:**

Strengths:

The idea proposed in this paper to address ambiguous instructions in GUI automation is interesting, and some practical attempts have also been made.


Weaknesses:

1. Although the motivation of privacy protection proposed in this paper is reasonable, the presented approach does not actually address this challenge. Using local/cloud models in combination can reduce some redundant information being uploaded to the cloud, but it cannot prevent the exposure of sensitive information. For example, in the scenario described in the Introduction, even if only the visual information of the search bar is uploaded to the cloud-based model, the search bar often contains recommendation content and may reveal user preferences to the cloud model. The “sufficient UI content” necessary for decision-making inevitably carries private information, which may contradict the paper’s claim of privacy protection.

2. The technical contribution of this paper is also limited. The multi-agent framework design is rather naive and has already been attempted and validated in many previous works [1,2].

3. I have doubts about the practical significance of the Reduction Rate metric. This metric only represents the quantitative reduction compared to uploading the entire UI to the cloud, but (1) the cloud model’s reasoning efficiency is unlikely to improve significantly because of it, and (2) it does not guarantee that the selected UI elements are free of sensitive information.

4. The paper lacks performance evaluation on the commonly used dataset AndroidWorld [3].

[1]. Wang J, Xu H, Jia H, et al. Mobile-agent-v2: Mobile device operation assistant with effective navigation via multi-agent collaboration[J]. arXiv preprint arXiv:2406.01014, 2024.

[2]. Agashe S, Han J, Gan S, et al. Agent s: An open agentic framework that uses computers like a human[J]. arXiv preprint arXiv:2410.08164, 2024.

[3] Rawles C, Clinckemaillie S, Chang Y, et al. Androidworld: A dynamic benchmarking environment for autonomous agents[J]. arXiv preprint arXiv:2405.14573, 2024.

---

> ### Author Rebuttal · Authors · 2025-07-31
>
> Thanks so much for your thoughtful comments. We've addressed all your concerns in detail below.
>
> > Q1: Preventing the exposure of sensitive information and practical significance of the Reduction Rate metric.
>
> A1: We agree with the reviewer that the "sufficient UI content" necessary for task completion may inevitably include private information. From the outset, **our design does not claim to eliminate all sensitive exposure**, but rather aims to **reduce the scope and amount of potential privacy leakage**. By selectively uploading only the most relevant UI blocks, our approach reduces **unnecessary privacy exposure**, thereby lowering the probability of **transmitting task-irrelevant sensitive content** to the cloud.
>
>
> We have also followed the suggestion to conduct a **quantitative privacy analysis on the sensitive information omitted by CORE**, specifically, comparing the sensitive elements uploaded to the cloud by the GPT-4o baseline and our method. We **categorize sensitive data into 8 categories**: (1) Identity & Account(e.g., username, profile); (2) Location & Schedule(e.g., home address, calendar events); (3) Contacts & Communication(e.g., contact information, messages, call logs); (4) Media & Files(e.g., file name, file content); (5) Device & Usage Info(e.g., device ID, storage); (6) Behavior & Preferences(e.g., interests, history, custom settings); (7) Finance & Security(e.g., payments, passwords, transactions); and (8) Other Sensitive Information. For each step in a task, we have analyzed all uploaded UI elements using **Qwen2.5-max** to **identify sensitive content and assign each sensitive UI element to one of the above categories**.
>
> **Results on DroidTask:**
> | |Identity & Account|Location & Schedule|Contacts & Communication|Media & Files|Device & Usage Info|Behavior & Preferences|Finance & Security|Other Sensitive Information|Total|
> | :------: | :------: | :------: |:------: |:------: |:------: | :------: |:------: | :------: | :------: |
> |GPT-4o|91| 226| 458| 147| 0| 45| 2| 0| 969|
> |Ours|50| 70 | 122| 32 | 0| 10| 2| 0| 286|
> |Reduction|45.05%|69.03%|73.36%|78.23%|/|77.78%|0.00%|/|70.49%|
>
> **Results on AndroidLab:**
> | |Identity & Account|Location & Schedule|Contacts & Communication|Media & Files|Device & Usage Info|Behavior & Preferences|Finance & Security|Other Sensitive Information|Total|
> | :------: | :------: | :------: |:------: |:------: |:------: | :------: |:------: | :------: | :------: |
> |GPT-4o|189 |299| 273| 12| 10| 77| 102| 1| 963|
> |Ours|111| 114| 203| 12| 5 |53 |90  |1  |589|
> |Reduction|41.27%|61.87%|25.64%|0.00%|50.00%|31.17%|11.76%|0.00%|38.84%|
>
> The results show that **our CORE significantly reduces sensitive UI exposure by 70.49% and 38.84% on DroidTask and AndroidLab, respectively.** These reductions **align with the overall element reduction rates of 55.60% and 34.96%** reported in the submission, indicating that **reducing overall UI exposure effectively lowers the risk of privacy leakage.**
>
>
> We have further **manually inspected and understood where the reductions come from**. The key findings are that most sensitive data fell into the **Location & Schedule** and **Contacts & Communication** categories due to apps like Calendar, Contacts, and Messenger. We show some detailed cases of our CORE as follows:
>
> * In a Calendar task (creating a new event), only the context relevant to the new event is uploaded, omitting previously scheduled events on the screen that reveal the user’s calendar.
> * In a Contacts task (updating a phone number), only the relevant phone number field is uploaded, while fields like email and birthday are excluded.
> * In a Messenger task (sending a message), only the current message block is uploaded to the cloud; prior chat history is not.
>
> The above results have demonstrated that **our CORE reduces unnecessary sensitive UI exposure, indeed increasing user comfort and trust in the mobile agent**.
>
> ---
>
> > Q2: Technical novelty of CORE, compared to previous multi-agent work [1,2].
>
> A2: We first clarify that prior work explored multi-agent framework design for the objective of **improving task accuracy**. In contrast, our CORE is developed for a different new objective: **minimizing unnecessary UI exposure to the cloud LLM without compromising decision quality**.
>
>
> The difference on the design objective also leads to the key difference on the design settings. Prior work typically **leveraged strong cloud LLMs with different prompts/roles [1, 2] and tools (e.g. retrieving external knowledge [2]), where they all can access the full UI**. In contrast, **our CORE coordinates between a cloud strong LLM and a local weak LLM**. In particular, **the powerful cloud LLM has strong reasoning ability but cannot access the full UI, whereas the weak local LLM has access to the full UI but has limited reasoning capacity**. Since the cloud LLM cannot access the full UI while also needs to identify which inputs are necessary for planning and decision-making, this strongly motivates **the collaboration between the cloud strong LLM and the local weak LLM for co-planning and co-decision-making**.
>
>
>
> The above new settings finally lead to the following design novelty in our CORE pipeline:
>
> * Our **co-planning stage and cloud–local LLMs collaboration in it** differ from typical task decomposition designs. Here, the **local weak LLM** does not attempt to directly determine the correct sub-task. Its **limited reasoning ability** makes this unreliable. Instead, it generates **UI-region-based sub-task candidates without concrete UI content**, which are then passed to the cloud strong LLM. **The cloud strong LLM**, leveraging its stronger reasoning ability and task history (but **without seeing the full UI**), **selects and refines the most appropriate sub-task**.
> * In the **co-decision-making stage**, unlike prior work that focuses solely on decision accuracy, we aim to **balance decision quality and UI reduction**. **The local LLM decides which UI blocks to upload**, as **the cloud LLM cannot identify them without direct UI access, but needs to make decision**. The validated sub-task from **co-planning** plays a key role in **helping the local LLM rank and select necessary blocks**. **The cloud LLM makes decisions using incomplete but sufficient UI blocks** provided by the local LLM.
>
> [1] Wang J, Xu H, Jia H, et al. Mobile-agent-v2: Mobile device operation assistant with effective navigation via multi-agent collaboration[J]. arXiv preprint arXiv:2406.01014, 2024.
>
> [2] Agashe S, Han J, Gan S, et al. Agent s: An open agentic framework that uses computers like a human[J]. arXiv preprint arXiv:2410.08164, 2024.
>
> ---
>
> > Q3: Evaluation on AndroidWorld.
>
> A3: We have followed the suggestion to **supplement the evaluation on AndroidWorld**. We clarify that our pipeline relies on the preprocessing of XML trees extracted via *uiautomator2*, which differs from the default UI extraction tool used by AndroidWorld’s *A11Y_FORWARDER_APP*. **The difference of XML tree extraction affects and requires the reconstruction of the default block partitioning module in our CORE**, which leverages the XML layout to group semantically related elements and faciliates better block filtering. Due to time limitation, we directly use the UI elements provided by AndroidWorld and evenly divide them into three blocks, while the subsequent two stages of CORE remain unchanged. **The evaluation results of our CORE with the naive even block partitioning** are shown as follows.
>
> | Method | Success Rate   | Reduction Rate |
> | :--: | :--: |  :--: |
> | Qwen2.5-Max (Baseline) | 35.34%| 0.00%          |
> | Ours with Even Block Partitioning (Qwen2.5-Max + Gemma-2-9B-Instruct) | 24.13%| 45.12%|
>
> We can observe that **our CORE framework achieves 45.12% of UI reduction, and the success rate decreased from 35.34% to 24.13%**. The drop in success rate is **partly attributable to the naive even block partitioning strategy and the inclusion of some very long-horizon and question-answering tasks in AndroidWorld**. As validated in our ablation study in Section 4.3, **replacing the XML-layout-informed partitioning with naive even partitioning can degrade task success rate (e.g., drop by 6.99% on DroidTask), but it allows us to test the feasibility of our overall framework. We will include more comprehensive results on AndroidWorld in the revision.**
>
> ---
>
> > Q4: Case study.
>
> A4: We **have included several practical case studies in Appendix E (Figures 1–5) of the supplementary material**, illustrating multi-step, cross-screen trajectories under our CORE. We agree with the reviewer that adding a **dedicated case study section in the main text** would greatly improve clarity. In the revision, we will highlight some typical cases to **qualitatively demonstrate our UI exposure reduction and the corresponding privacy gains**, some of which are briefly illustrated in **our response A1 to your comment Q1**.

---

> > ### Comment · Reviewer_PSyJ · 2025-08-05
> > **Response by Reviewer PSyJ**
> >
> > Thank you very much for your detailed and comprehensive response. While the paper does not completely solve the privacy exposure issue in GUI automation, it is an insightful and exploratory work. Based on the quantitative privacy analysis provided by the authors, the proposed method demonstrates a meaningful reduction in privacy exposure. Considering the feedback from other reviewers, I will raise my score to 4.

---

> > > ### Author Response · Authors · 2025-08-05
> > >
> > > Thank you very much for your thoughtful and encouraging feedback. We sincerely appreciate your recognition of our work as both insightful and exploratory, and we’re grateful for your positive evaluation of our efforts to reduce privacy exposure in mobile GUI automation. Your suggestion to incorporate both quantitative and qualitative analyses of privacy reduction was particularly valuable, and we’re pleased that the results we presented were found meaningful.
> > >
> > > We would also like to respectfully clarify that while our current design does not aim to eliminate all sensitive exposure, it focuses on significantly filtering out UI elements irrelevant to the user's task to minimize unnecessary privacy leakage. Looking ahead, we hope this work guides future research in mobile agents to not only emphasize task accuracy but also to prioritize reducing unnecessary UI exposure.

---

### Official Review · Reviewer_p2hj · 2025-06-30

**Clarity:** 3
**Significance:** 2
**Originality:** 2
**Rating:** 4
**Confidence:** 3

**Summary:**

The paper proposes CORE, a framework to addresses the problem of excess UI data being sent to cloud LLMs by mobile agents. The authors provide empirical results to show that CORE can reduce the number of UI elements uploaded by over 50% while keeping task success rates within ~5% of a full-access cloud agent.

**Questions:**

1.	Have you considered running a small-scale user study to measure whether hiding non-essential UI blocks actually increases users’ comfort or trust in the agent?
2.	Can you provide a more concrete analysis of what sensitive information is omitted by CORE?
3.	Aside from the prompt templates, what are the differences between your LLM models collaboration pipeline, which comprises Co-planning and Co-decision-making, and those employed in prior works [1,2,3] ?

[1] Dai G, Jiang S, Cao T, et al. Advancing mobile gui agents: A verifier-driven approach to practical deployment[J]. arXiv preprint arXiv:2503.15937, 2025.
[2] Christianos F, Papoudakis G, Coste T, et al. Lightweight neural app control[J]. arXiv preprint arXiv:2410.17883, 2024.
[3] Mei K, Zhu X, Xu W, et al. Aios: Llm agent operating system[J]. arXiv preprint arXiv:2403.16971, 2024.

**Ethical Concerns:**

["NO or VERY MINOR ethics concerns only"]

**Final Justification:**

While the paper's empirical results are comprehensive, its novelty remains somewhat limited.

**Limitations:**

Yes

**Quality:**

2

**Strengths And Weaknesses:**

Strengths:
1)	The paper focuses on an important and timely research topic, namely, the LLM agents for smartphone operation.
2)	The paper is easy to follow.
3)	The GUI exposure problem is a novel problem to solve.

Weaknesses:
1)	The practical value of the proposed method is insufficiently justified. The paper’s primary motivation is reducing data exposure. However, the experimental results (Table 1 and Figure 4) show that CORE incurs higher latency and lower accuracy than the cloud-only baselines. To strengthen the case for CORE, the authors should demonstrate that limiting GUI exposure yields tangible privacy or usability gains. For example, they could: 1) include a user study measuring perceived comfort or trust when sensitive fields are hidden; or 2) include quantitative privacy-risk analysis (e.g., metrics on sensitive information leakage).
2)	The novelty of the CORE framework is incremental. While CORE’s hierarchical local/cloud architecture is well motivated, similar two-stage frameworks have appeared recently (e.g., [1, 2, 3]). The manuscript needs to clearly articulate what distinguishes CORE’s “co-planning/co-decision” mechanism from prior work.
[1] Dai G, Jiang S, Cao T, et al. Advancing mobile gui agents: A verifier-driven approach to practical deployment[J]. arXiv preprint arXiv:2503.15937, 2025.
[2] Christianos F, Papoudakis G, Coste T, et al. Lightweight neural app control[J]. arXiv preprint arXiv:2410.17883, 2024.
[3] Mei K, Zhu X, Xu W, et al. Aios: Llm agent operating system[J]. arXiv preprint arXiv:2403.16971, 2024.

---

> ### Author Rebuttal · Authors · 2025-07-31
>
> Thanks for your thorough comments. We have provided detailed responses to your key concerns as follows.
>
> > Q1: Quantitative privacy gains and user study for perceived comfort or trust.
>
> A1: We have followed the suggestion to conduct a **quantitative privacy analysis on the sensitive information omitted by CORE**, specifically, comparing the sensitive elements uploaded to the cloud by the GPT-4o baseline and our method. We **categorize sensitive data into 8 categories**: (1) Identity & Account(e.g., username, profile); (2) Location & Schedule(e.g., home address, calendar events); (3) Contacts & Communication(e.g., contact information, messages, call logs); (4) Media & Files(e.g., file name, file content); (5) Device & Usage Info(e.g., device ID, storage); (6) Behavior & Preferences(e.g., interests, history, custom settings); (7) Finance & Security(e.g., payments, passwords, transactions); and (8) Other Sensitive Information. For each step in a task, we have analyzed all uploaded UI elements using **Qwen2.5-max** to **identify sensitive content and assign each sensitive UI element to one of the above categories**.
>
> **Results on DroidTask:**
> | |Identity & Account|Location & Schedule|Contacts & Communication|Media & Files|Device & Usage Info|Behavior & Preferences|Finance & Security|Other Sensitive Information|Total|
> | :------: | :------: | :------: |:------: |:------: |:------: | :------: |:------: | :------: | :------: |
> |GPT-4o|91| 226| 458| 147| 0| 45| 2| 0| 969|
> |Ours|50| 70 | 122| 32 | 0| 10| 2| 0| 286|
> |Reduction|45.05%|69.03%|73.36%|78.23%|/|77.78%|0.00%|/|70.49%|
>
> **Results on AndroidLab:**
> | |Identity & Account|Location & Schedule|Contacts & Communication|Media & Files|Device & Usage Info|Behavior & Preferences|Finance & Security|Other Sensitive Information|Total|
> | :------: | :------: | :------: |:------: |:------: |:------: | :------: |:------: | :------: | :------: |
> |GPT-4o|189 |299| 273| 12| 10| 77| 102| 1| 963|
> |Ours|111| 114| 203| 12| 5 |53 |90  |1  |589|
> |Reduction|41.27%|61.87%|25.64%|0.00%|50.00%|31.17%|11.76%|0.00%|38.84%|
>
> The results show that **our CORE significantly reduces sensitive UI exposure by 70.49% and 38.84% on DroidTask and AndroidLab, respectively.** These reductions **align with the overall element reduction rates of 55.60% and 34.96%** reported in the submission, indicating that **reducing overall UI exposure effectively lowers the risk of privacy leakage.**
>
> We have further **manually inspected and understood where the reductions come from**. The key findings are that most sensitive data fell into the **Location & Schedule** and **Contacts & Communication** categories due to apps like Calendar, Contacts, and Messenger. We show some detailed cases of our CORE as follows:
>
> * In a Calendar task (creating a new event), only the context relevant to the new event is uploaded, omitting previously scheduled events on the screen that reveal the user’s calendar.
> * In a Contacts task (updating a phone number), only the relevant phone number field is uploaded, while fields like email and birthday are excluded.
> * In a Messenger task (sending a message), only the current message block is uploaded to the cloud; prior chat history is not.
>
> The above results have demonstrated that **our CORE reduces unnecessary sensitive UI exposure, indeed increasing user comfort and trust in the mobile agent**.
>
> ---
>
> > Q2: Technical novelty of CORE compared with prior work [1,2,3].
>
> A2: We first clarify that **prior work** explored two-stage designs for the objective of **improving task accuracy**. In contrast, our CORE is developed for a different new objective: **minimizing unnecessary UI exposure to the cloud LLM without compromising decision quality**.
>
>
> The difference on the design objective also leads to the key difference on the design settings. **Prior work** typically **leveraged strong cloud LLMs with different prompts/roles [1], tools [3], or designed specialized models [2], where they all can access the full UI**. In contrast, **our CORE coordinates between a cloud strong LLM and a local weak LLM**. In particular, **the powerful cloud LLM has strong reasoning ability but cannot access the full UI, whereas the weak local LLM has access to the full UI but has limited reasoning capacity**. Since the cloud LLM cannot access the full UI while also needs to identify which inputs are necessary for planning and decision-making, this strongly motivates **the collaboration between the cloud strong LLM and the local weak LLM for co-planning and co decision-making**.
>
> The above new settings finally lead to the following design novelty in our CORE pipeline:
>
> * Our **co-planning stage and cloud–local LLMs collaboration in it** differ from typical task decomposition designs. Here, the **local weak LLM** does not attempt to directly determine the correct sub-task. Its **limited reasoning ability** makes this unreliable. Instead, it generates **UI region-based sub-task candidates without concrete UI content**, which are then passed to the cloud strong LLM. **The cloud strong LLM**, leveraging its stronger reasoning ability and task history (but **without seeing the full UI**), **selects and refines the most appropriate sub-task**.
> * In the **co-decision-making stage**, unlike prior work that focuses solely on decision accuracy, we aim to **balance decision quality and UI reduction**. **The local LLM decides which UI blocks to upload**, as **the cloud LLM cannot identify them without direct UI access, but needs to make decision**. The validated sub-task from **co-planning** plays a key role in **helping the local LLM rank and select necessary blocks**. **The cloud LLM makes decisions using incomplete but sufficient UI blocks** provided by the local LLM.
>
> [1] Dai G, Jiang S, Cao T, et al. Advancing mobile gui agents: A verifier-driven approach to practical deployment[J]. arXiv preprint arXiv:2503.15937, 2025.
>
> [2] Christianos F, Papoudakis G, Coste T, et al. Lightweight neural app control[J]. arXiv preprint arXiv:2410.17883, 2024.
>
> [3] Mei K, Zhu X, Xu W, et al. Aios: Llm agent operating system[J]. arXiv preprint arXiv:2403.16971, 2024.

---

> > ### Comment · Reviewer_p2hj · 2025-08-05
> >
> > The authors' response has addressed some of my concerns. While the paper's empirical results are comprehensive, its novelty remains somewhat limited. Accordingly, I will slightly increase my score.

---

> > > ### Author Response · Authors · 2025-08-05
> > >
> > > Thank you very much for your thoughtful comments and encouraging feedback on the improved score. We’re pleased that our response has addressed most of your key concerns. We truly appreciate that you found our empirical results comprehensive.
> > >
> > > Regarding the remaining concern about novelty, we would like to respectfully clarify that the unique contributions of our work arise from a newly defined problem and objective, as well as a novel asymmetric multi-agent collaboration framework. In particular, we identify a brand **new problem in mobile agents with a distinct objective**: **minimizing unnecessary task-irrelevant UI exposure to cloud LLMs**, which fundamentally differs from **existing work** that **primarily focused on optimizing task performance**. This new problem naturally leads to a **novel multi-agent collaboration paradigm: a strong cloud LLM without UI access collaborating with a weak local LLM that has full UI access but limited reasoning capabilities.** **The asymmetry on LLM capability and UI access** further motivates **a new co-planning and co-decision framework, where strong and weak agents coordinate by leveraging their complementary strengths under their respective constraints.** In addition, the preprocessing stage also introduces a novel block partitioning algorithm driven by the hierarchical structure of the UI to support the collaborative framework. To the best of our knowledge, this kind of asymmetric multi-agent coordination, especially under UI access constraints, was not explored in **existing agent work, which leveraged strong cloud LLMs with different prompts/roles, tools, and/or designed specialized models, where they all can access the full UI.**
> > >
> > > We hope our work opens up **new avenues for future research in mobile agents**, encouraging the community to consider **not only task performance but also principles such as minimizing UI exposure**, which can be critical for **enhancing privacy and improving real-world user trust and comfort**.

---

### Official Review · Reviewer_ZB4i · 2025-07-05

**Clarity:** 3
**Significance:** 3
**Originality:** 3
**Rating:** 4
**Confidence:** 3

**Summary:**

The paper presents CORE, a framework that blends cloud and local large language models (LLMs) to streamline task automation on mobile devices while cutting down on unnecessary UI data sent to the cloud. By using XML-based layout-aware block partitioning, collaborative planning, and decision-making, CORE reduces UI exposure by up to 55.6% while keeping task success rates close to cloud-only models. Tests on DroidTask and AndroidLab datasets show it maintains strong performance—within 5% of GPT-4o—while significantly outperforming local-only models by up to 46.85%.

**Questions:**

1. How does CORE perform on tasks requiring navigation across multiple screens or highly dynamic UIs, like social media apps? Testing these could strengthen claims of robustness, and strong results here would boost my confidence in its general applicability.
2. Can you quantify the local LLM’s computational burden, especially on low-end devices? A breakdown of resource demands (e.g., memory, CPU) would clarify feasibility and could address my concerns about practical deployment.
3. Why focus solely on XML-based UIs? Exploring compatibility with screenshot-based or multimodal approaches could broaden CORE’s relevance. Showing adaptability to other formats would elevate the paper’s impact.

**Ethical Concerns:**

["NO or VERY MINOR ethics concerns only"]

**Final Justification:**

Thank you to the authors for their thoughtful and insightful responses. I maintain my positive score, and I hope this paper will be accepted.

**Limitations:**

yes

**Quality:**

3

**Strengths And Weaknesses:**

Strengths:
- The paper tackles a real privacy concern in mobile agents by reducing UI data uploads, a practical issue given smartphones’ sensitive nature. The proposed CORE framework smartly leverages both local and cloud LLMs to balance privacy and performance, which feels like a meaningful step forward.
- Experiments are thorough, covering two datasets (DroidTask and AndroidLab) with diverse apps and tasks, and results are compelling—55.6% UI reduction with minimal accuracy loss is impressive. The ablation study nicely pinpoints the value of each component, like co-planning and multi-round accumulation.
- The writing is engaging and clear, with figures like the CORE pipeline and XML partitioning visuals making the technical details digestible. The problem formulation is precise, grounding the work in a well-defined optimization goal.
- The idea of using XML structure for block partitioning is clever, preserving the UI’s logical design in a way that feels intuitive and practical for mobile interfaces.

Weaknesses:
- The evaluation is solid but could dig deeper into edge cases, like apps with highly dynamic UIs or tasks needing multiple steps across screens. It’s unclear how CORE handles these trickier scenarios, which could limit its real-world robustness.
- The paper glosses over the computational cost of local LLM processing. While cloud token usage is discussed, the added latency from local models on resource-constrained devices feels underexplored, especially for consumer-grade phones.
- The reliance on XML-based UI representations might not generalize to screenshot-based or multimodal agents, which are increasingly common. This narrows the framework’s immediate applicability.

---

> ### Author Rebuttal · Authors · 2025-07-31
>
> We sincerely thank the reviewer's insightful and valuable comments. We have provided responses to address all your concerns as follows.
>
> > Q1: Performance on tasks requiring multiple steps across multiple screens or highly dynamic UIs, like social media apps.
>
> A1: **Regarding multi-step, multi-screen tasks**, we want to clarify that **both benchmark datasets used in our evaluation (DroidTask and AndroidLab) are composed of multi-step, multi-screen tasks**:
>
> * In **DroidTask**, according to **Figure 6(a) in its paper [1]**, human trajectories span an average of **5 screens**, with some tasks involving over **10 screens**.
> * In **AndroidLab**, as shown in **Figure 5(f) of its paper [2]**, tasks are even more complex, requiring on average **7 screen transitions**, with some tasks over **10 screens**.
>
> We **have also included several practical case studies in Appendix E (Figures 1–5) of the supplementary material**, illustrating multi-step, cross-screen trajectories under our CORE.
>
> **Regarding social media apps with highly dynamic UIs**, we have followed your suggestion to add experiments on a subset of the **LlamaTouch** dataset [3], comprising tasks from common social media apps. We tested 64 tasks across **Instagram, X (Twitter), Reddit, and Pinterest**, with the following results:
>
> | Method | Success Rate   | Reduction Rate |
> | :--: | :--: |  :--: |
> | GPT-4o (Baseline) | 70.31% (45/64) | 0.00%          |
> | Ours (GPT-4o + Gemma-2-9B-Instruct) | 60.94% (39/64) |  48.94%|
>
> Evaluation results show that **our CORE completed only 6 fewer tasks than the full-UI GPT-4o baseline, while reducing UI exposure by 48.94%**, revealing effectiveness on social media apps with highly dynamic UIs.
>
> [1] Wen, Hao, et al. "Autodroid: Llm-powered task automation in android." Proceedings of the 30th Annual International Conference on Mobile Computing and Networking. 2024.
>
> [2] Xu, Yifan, et al. "Androidlab: Training and systematic benchmarking of android autonomous agents." arXiv preprint arXiv:2410.24024 (2024).
>
> [3] Zhang, Li, et al. "Llamatouch: A faithful and scalable testbed for mobile ui task automation." Proceedings of the 37th Annual ACM Symposium on User Interface Software and Technology. 2024.
>
> ---
>
> > Q2: Local LLM’s overhead and resource demands on low-end devices.
>
> A2: We have followed your suggestion to deploy a quantized **Qwen2.5-7B-Instruct** that has been used in our evaluation on a **Xiaomi smartphone** using **Alibaba’s mobile inference engine MNN**. We select all 5 tasks in Applauncher app from DroidTask dataset for overhead testing. **The smartphone's hardware configurations, the inference latency, the CPU usage, and the memory usage per task** are shown below.
>
> | Device        | DRAM | SoC | CPU  |
> | :---: | :---: | :---: | :---: |
> | Xiaomi 15 Pro | 16GB |  Qualcomm Snapdragon  8 Elite   | 2\*Oryon(4.32GHz) + 6\*Oryon(3.53GHz) |
>
> | Model|Prefill Speed(token/s) |Decode Speed (token/s)|Prefill Time per Query(s)|Decode Time per Query(s)|Prefill Time per Task(s)|Decode Time per Task(s)|Memory Usage(GB)|CPU Utilization(%)|
> | :---: | :---: | :---: | :---: | :---: |:---: | :---: |:---: | :---: |
> |Qwen2.5-7B-Instruct|49.06|6.52|5.28|4.65|75.05|65.99|6.9|770|
>
> We can observe that **the inference latency of on-device LLM is roughly 140s per task, the CPU roughly occupies 8 cores, and the memory usage is below 7GB**.
>
> We further have adopted more universal LLM engine **llama.cpp** to deploy all three local LLMs used in our evaluation for overhead testing, including **Qwen2.5-7B-Instruct, Llama-3.1-8B-Instruct, and Gemma-2-9B-Instruct**. We find that **the latency of Qwen2.5-7B-Instruct using llama.cpp is about 5× higher than MNN**, and **the CPU and memory usage of llama.cpp are comparable to MNN**. In addition, **the overhead increases slightly with the model size increasing from 7B to 9B.**
>
> We finally want to clarify that **efficient LLM inference on mobile devices is still a great challenge in existing mobile agent work**, primarily due to limited hardware resources. As a result, **existing work typically called cloud LLM APIs or ran LLMs on more powerful computers and servers**. We initially followed this common practice using a consumer-grade RTX 4090D GPU for overhead testing with the results of Qwen2.5-7B-Instruct shown below. We can find that **the inference speed on 4090D** is **4.39× faster** than using MNN on **Xiaomi 15 Pro**.
>
> |Methods| Cloud Latency per Task (s) | Local Latency per Task (s) | Total Latency per Task (s) | (Total Latency) Ratio vs GPT-4o Baseline |
> | :------: | :------: | :------: | :------: | :------: |
> | GPT-4o (Baseline) | 40.32| 0.00|40.32|1.00x|
> | Ours (GPT-4o + local LLM on RTX 4090D)| 29.15| 32.12| 61.27| 1.52x|
> | Ours (GPT-4o + local LLM on Smartphone)| 29.15| 141.04| 170.19| 4.22x|
>
> ---
>
> > Q3: Adaptability to screenshot-based or multimodal settings.
>
> A3: We first clarify that we initially chose to build CORE on XML-based UI representations because, **prior to 2025**, foundation multimodal LLMs lacked sufficient reliability for accurate UI element localization. In constrat, **XML provided precise bounding box information** and a structured hierarchy that is essential for accurate decision-making.
>
> We also agree with the reviewer that multi-modality is important, as **XML and screenshots each have complementary advantages**. Screenshots offer richer visual context, while XML encodes semantic metadata (e.g., content-description), reflecting developer intent.
>
> In fact, **our CORE framework is modality-agnostic** and can be **conveniently adapted to incorporate screenshots**. This is because our key contribution lies in proposing a general workflow designed to address a new problem: reducing unnecessary UI exposure to the cloud LLM while preserving decision quality. All the key modules of **layout-aware block partitioning, co-planning, and co-decision-making** are **all modality-agnostic**. **Each module can operate on XML-only, screenshot-only, or combined inputs**. For example, the block partitioning module can use either an XML-based strategy or a visual segmentation approach. Similarly, the local and cloud LLMs can be replaced with multimodal LLMs. The rest of the collaborative pipeline remains unchanged.
>
> To extend CORE to multimodal input, we have adopted a straightforward masking strategy: CORE’s selective UI reduction enables us to precisely identify which UI elements should be masked. We **capture screenshots of the UI and apply gray masks to the bounding boxes of elements that CORE removes, generating a masked image. This masked screenshot is then passed to a cloud multimodal LLM (e.g., GPT-4o in our evaluation)** for decision-making, alongside the textual UI element descriptions used in the original XML-based setting. The GPT-4o baseline receives the full screenshot, while our CORE uses the masked one.
>
> We show the evaluation results on DroidTask and AndroidLab as follows.
>
> **Results on DroidTask:**
> |Methods (+screenshot)|Success Rate|Reduction Rate|Cloud Latency per Task (s)|Local Latency per Task (s) | Total Latency per Task (s) |
> | :------: | :------: | :------: |:------: |:------: |:------: |
> |GPT-4o (Baseline)| 74.13% | 0.00% | 42.19 | 0.00 | 42.19 |
> |Ours (GPT-4o + Gemma-2-9B-Instruct)| 69.93% | 56.84% | 33.81 | 30.98 | 64.79 |
>
> **Results on  AndroidLab:**
> |Methods (+screenshot)|Success Rate|Reduction Rate|Cloud Latency per Task (s)|Local Latency per Task (s) | Total Latency per Task (s) |
> | :------: | :------: | :------: |:------: |:------: |:------: |
> |GPT-4o (Baseline)| 46.94% | 0.00% | 46.33 | 0.00 | 46.33 |
> |Ours (GPT-4o + Gemma-2-9B-Instruct)| 39.80% | 37.51% | 38.75 | 38.44 | 77.19 |
>
> These results show that **CORE remains effective with multimodal input**: it achieves success rates close to the GPT-4o baseline while reducing UI exposure by **56.84%** and **37.51%** on DroidTask and AndroidLab, respectively.

---

> > ### Comment · Reviewer_ZB4i · 2025-08-05
> > **Reviewer Response**
> >
> > Thank you to the authors for the detailed response, which has addressed all of my concerns. This is a good paper, and I will maintain my positive score.

---

> > > ### Author Response · Authors · 2025-08-05
> > >
> > > We sincerely thank you for recognizing the contribution and value of our work. We're grateful for your thoughtful review and are pleased that our responses have addressed all your concerns. We will incorporate the suggested revisions into the final version as recommended.

---

### Note · Authors · 2025-08-12

Dear Reviewers, ACs, and SACs,

We sincerely thank all the reviewers for the constructive feedback and thoughtful engagement. We are pleased that our rebuttal has addressed the concerns  raised, and we greatly appreciate all your positive evaluations and strong support. In particular, we are truly grateful for the reviewers’ acknowledgment of the following highlights and key contributions of our work:

- The importance of reducing unnecessary task-irrelevant UI exposure to cloud LLMs in mobile agents, and the insightful, exploratory character of our work as the first to identify, formulate, and address this problem. (Reviewers meZ1, PSyJ, p2hj, ZB4i)

- The technical novelty of our asymmetric multi-agent collaboration framework, where a strong cloud LLM without UI access collaborates with a weak local LLM that has full UI access but limited reasoning capabilities, coordinating by leveraging their complementary strengths under their respective constraints. (Reviewers meZ1, ZB4i)

- The comprehensive experimental evaluation demonstrating our framework’s practical effectiveness, further strengthened by demonstrating not only overall UI exposure reduction but also substantial mitigation of privacy exposure. (Reviewers meZ1, PSyJ, p2hj, ZB4i)

- The paper is well-written, clear, and easy to follow. (Reviewers p2hj, ZB4i)

We once again express our sincere appreciation for your generous time, dedicated effort, and constructive feedback. Your insights have been invaluable in refining our work, and we are truly grateful for your thoughtful engagement throughout the review process.

Best regards,

Authors of Paper #11810

---

### Decision · Program_Chairs · 2025-09-17

**Decision:**

Accept (poster)

**Comment:**

This paper present a strategy to collaborate local and cloud LLM to minimize UI exposure in mobile agents. Through the discussion in Rebuttal phase, the reviewers reach consensus on accepting this paper.